



# The Impact of Background ENSO and NAO Conditions and Anomalies on the Modeled Response to Pinatubo-Sized Volcanic Forcing

Helen Weierbach[1,2,3], Allegra N. LeGrande[4,5], and Kostas Tsigaridis[5,4]

[1]Earth and Environmental Sciences Area, Lawrence Berkeley National Laboratory, Berkeley, CA, USA
[2]Tulane University, New Orleans, LA, USA
[3]Lamont Doherty Earth Observatory, Columbia University, New York, NY, USA
[4]NASA Goddard Institute for Space Studies, New York, NY, USA
[5]Center for Climate Systems Research, Columbia University, New York, NY, USA

**Correspondence:** Kostas Tsigaridis (kostas.tsigaridis@columbia.edu)

**Abstract.** Strong, strato-volcanic eruptions are a substantial, intermittent source of natural climate variability. Background atmospheric and oceanic conditions such as El Niño Southern Oscillation (ENSO) and the North Atlantic Oscillation (NAO) also naturally impact climate on regular time scales. We examine how background conditions of ENSO and NAO impact the climate's response to a Pinatubo-type eruption using a large (81-member) ensemble of model simulations in GISS Model E2.1-
G. Simulations are sampled from possible background conditions under the protocol of the coordinated CMIP6 Volcanic Model Intercomparison Project (VolMIP) – where aerosols are forced with respect to time, latitude, and height. We analyze paired anomalous variations (perturbed - control) to understand changes in global and regional climate responses under positive, negative, and neutral ENSO and NAO conditions. In particular, we find that for paired anomalies there is a high probability of strong ($\sim 1.5$ °C) post-eruptive winter warming for negative NAO ensembles with analysis coincident with decreased
lower stratospheric temperature at the poles, decreased geopotential height, and strengthening of the stratospheric polar vortex. Historical anomalies (relative to climatology) show no mean warming and suggest that the strength of this response is impacted by control conditions. Again using paired anomalies, we also observe that positive and negative ENSO ensembles relax the ENSO anomaly in the first post-eruptive Boreal Winter while neutral-phase ensembles are variable and show no clear response. In general, paired anomalies give insight into the evolution of the climate response to volcanic forcing, but are significantly
impacted by background climate conditions present in control conditions.

## 1 Introduction

Strong, explosive volcanic eruptions are an intermittent, natural source of climate variability acting on both inter-annual and decadal scales. Explosive volcanic eruptions eject sulfur dioxide, halogens, ash and water vapor into the stratosphere, where the particles are converted into sulfate aerosols (LeGrande et al., 2016). The loading of stratospheric aerosols increases aerosol
optical depth of the atmosphere (Lacis, 2015), thus imposing a radiative forcing via scattering of shortwave radiation and absorption of longwave radiation in the stratosphere (Zanchettin et al., 2013).



The impact that a strong volcanic eruption makes on the climate system depends on many factors including size, ejection height, and location of the eruption. Timing of an eruption also contributes to the climate system's response, with seasonal timing and background climate conditions at the time of the eruption also influencing the climate response. These factors impact the amount, location and dynamics of how aerosols are loaded in the atmosphere, significantly impacting how the climate system responds to the perturbation. Mt. Pinatubo is an example of one such strong volcanic eruption which erupted in the Philippines in June 1991, ejecting 18 Tg of $SO_2$ into the atmosphere at a height of 20 km (Stenchikov et al., 1998). The Mt. Pinatubo eruption has been widely studied as one of the largest volcanic eruptions in the last decade McCormick et al. (1995); Bluth et al. (1992); Stenchikov et al. (1998). Climate models are frequently used to study this eruption, as aerosols are relatively well constrained using satellite observations of the eruption. In this study, we analyze the modelled response to a Mt.Pinatubo sized eruption on Earth in the absence of greenhouse gases to determine the role of background conditions in the climate response to such a volcanic eruption.

The impact of a Pinatubo sized eruption on the Earth's climate system is significant; previous work has shown that Pinatubo-sized volcanic eruptions decrease radiative flux in the region [40 °N-40 °S] by around -4.3 W m$^{-2}$ at their peak aerosol forcing (Minnis et al., 1993), with radiative effects lasting for about two years after the eruption. In comparison, anthropogenic radiative forcing is estimated to have increased global energy budget by 2.3 W m$^{-2}$ over the industrial period (Myhre et al., 2013), making volcanic forcing a short-lived but substantial source of natural climate variability. The resulting impacts in the climate system, however, last years after volcanic aerosols have been depleted. The direct impacts of volcanic aerosols include cooling of the Earth's surface and warming of the stratosphere (Lacis, 2015). These direct impacts initiate many other changes in the climate system including changes in atmospheric circulation, the hydrological cycle, the cryosphere, and carbon cycle (Zanchettin et al., 2013).

## 1.1 Background Conditions and Volcanic Eruptions

Background climate conditions such as the El Niño Southern Oscillation System (ENSO) and North Atlantic Oscillation (NAO) continuously cause variations in Earth's climate over time (Philander, 1983; Allan et al., 1996; Timmermann et al., 2018). Like volcanic eruptions, they cause changes in regional and global climate on interannual time scales. When background conditions are combined with a strong volcanic perturbation, they can lead to different response pathways as seen in climate model simulations (Zanchettin et al., 2013). Thus, background climate conditions contribute significantly to variation in volcanic model simulations and uncertainty in climate predictions. (Zanchettin et al., 2013) examined this effect for a Tambora-sized volcanic eruption, finding that while the radiative forcing remained the same, different background conditions caused substantial variability in surface atmospheric and oceanic conditions.

Here we focus on how background ENSO and NAO conditions create variability in the response to a Pinatubo-sized eruption. In particular, we investigate how these background conditions cause changes in the evolution of the ENSO cycle and the northern hemisphere's first winter. These two responses have been studied through observational and model-based studies in the past decade with results varying in the magnitude and direction of the ENSO and NAO response. Few studies, however, have used large ensembles to investigate how these background conditions affect the modelled climate response. We find that





background atmospheric states (i.e. what state of NAO the climate system would normally be in) significantly impacts the significance of the winter warming response, but does not cause significant changes in the ENSO response in the GISS Model E2.1.

### 1.1.1 ENSO Response

El Niño Southern Oscillation (ENSO) is an important mode of climate variability which oscillates between positive (El Niño), neutral and negative (La Niña) phases at time scales of about 2-7 years in the equatorial Pacific Ocean (Predybaylo et al., 2017). During positive (negative) phases, the equatorial pacific experiences higher (lower) than average sea surface temperature anomalies. These oceanic changes initiate changes in both regional climate and global climate connections. Both observational (direct and proxy-based) and model-based studies have been used to examine how ENSO responds to large volcanic perturba-
tions. Some proxy-based and several modelling studies suggest that large, tropical volcanic eruptions increase the likelihood of an El Niño like anomaly following the eruption (Adams et al., 2003; Predybaylo et al., 2017; Khodri et al., 2017). This response is suggested to be particularly robust when the eruption occurs in the Northern Hemisphere due to the eruption shifting the ITCZ southward, thus weakening trade winds in the Tropical Pacific (Pausata et al., 2020). Weakened trade winds then cause El Niño like conditions via the Bjerknes feedback (Bjerknes, 1969).

Research has also focused on understanding the dynamics of the El Niño anomaly.One suggested mechanism is the ocean dynamical thermostat (Clement et al., 1996), a mechanism which is suggested to cause advection of warm water through differential cooling. A second hypothesis for a post-eruptive El Niño anomaly is post-eruptive land cooling over tropical Africa which initiates warming through the perturbation of Walker circulation cells (Khodri et al., 2017). Predybaylo et al. (2017) additionally studied the robustness of the simulated El Niño anomaly under varying background conditions at the time of
volcanic eruptions. Their modelling study showed enhanced El Niño like warming for all simulations except those where eruptions occurred in La Niña years (Predybaylo et al., 2017).

Despite several studies supporting El Niño like anomalies, still other observational and modelling studies suggest that there is no statistically significant El Niño like response after several large volcanic eruptions (Dee et al., 2020). These studies argue that anomalies found in observational records and model simulations are not statistically significant, and are rather within the
range of natural climate variability (Dee et al., 2020). Here we examine the post-eruptive ENSO response with GISS Model E2.1-G under varying background conditions of ENSO to determine if the model supports an El Niño like response after volcanic eruptions and or either of the proposed mechanisms.

### 1.1.2 Northern Hemisphere Winter Response

The northern hemisphere (NH) experiences a unique response during the first winter after large volcanic eruptions. Many obser-
vational (Graf et al., 2007; Christiansen, 2008) and modelling (Timmreck, 2012; Stenchikov et al., 2002) studies have supported a strengthening of the polar vortex the first winter after a large volcanic eruption. However, the robustness of this modelled response has been questioned. For example, CMIP5 models show variation in the prevalence of this response (Timmreck et al.,





2016; Driscoll et al., 2012) suggesting that large numbers of ensembles may be required to see a significant strengthening of the polar vortex (Bittner et al., 2016).

Some studies suggest that the simulated winter warming response in a model is within the range of internal variability (Polvani et al., 2019) and thus is not a robust response to volcanic eruptions. Increased polar vortex circulation is closely associated with an enhanced phase of the Arctic Oscillation(AO) and North Atlantic Oscillation (NAO). These two modes of natural climate variability are separately defined, but closely related in their associate climate impacts including surface temperature patterns (Cohen and Barlow, 2005). Thus, studies have often suggested that volcanic forcing causes an excited

positive phase of the AO and NAO (Christiansen, 2008; Shindell et al., 2004). These changes in circulation are hypothesized to lead to observed increases in surface temperature as seen the winter after the Pinatubo eruption (Robock and Mao, 1995; Kelly et al., 1996) common in positive phases of the AO and NAO. This increase in surface temperature is also seen in both observational and global modelling studies (Robock and Mao, 1992; Graft et al., 1993).

    The cause of an observed strengthening polar vortex circulation and winter warming response has also been addressed. One

widely cited cause of winter warming is differential heating of the stratosphere between the tropics and polar regions. When a tropical volcano erupts, the loading of aerosols causes stratospheric warming more strongly at tropical latitudes than high latitudes, termed the equator-to-pole temperature difference (Robock, 2000). This differential heating perturbs atmospheric circulation patterns leading to a strengthened stratospheric polar vortex which propagates downward into the troposphere through coupling between the stratosphere and troposphere. This downward propagation of increased westerly winds shifts the

tropospheric jet northward. Shifting of the jet in turn prevents cool, polar air from moving south of the pole, bringing warmer than average temperatures to high latitudes (Robock and Mao, 1995) indicative of winter warming.

    Others suggest that a strengthened stratospheric equator-to-pole gradient alone can not explain the observed response. Other factors, such as ozone depletion in the stratosphere (Stenchikov et al., 2002) have been found to also contribute to the enhanced circulation of the polar vortex. The connection between increased polar vortex circulation and winter warming has also been

questioned. Polvani et al. (2019) found no significant correlation between increased stratospheric polar vortex circulation and Eurasian surface temperature using ensembles of WACCM4 Pinatubo simulations, suggesting that winter warming is withing expected internal variability rather than a result of a dynamic response to volcanic aerosols. Other studies such as Driscoll et al. (2012) and Stenchikov et al. (2006) also find no consistent warming in the northern hemisphere, or strengthening of the polar vortex. Here we also evaluate the robustness and cause of the NH winter response in GISS E2.1-G.

**1.2 VolMIP**

To investigate how background conditions could play into the ENSO and NH winter responses, we run uniform simulations under different background conditions as defined by the Volcanic Model Intercomparison Project. The Volcanic Model Intercomparison Project (VolMIP) is part of the coordinated effort within the Model Intercomparison Project of CMIP6 (Eyring et al., 2016) that seeks to assess which climate responses to volcanic eruptions are robustly simulated in state of the art climate

models. VolMIP proposes a set of experiments each aiming to systematically quantify the modelled climate response to specific types of volcanic eruptions under a unified methodology to reduce variability between model studies. The 'volc-pinatubo-full'





VolMIP experiment addresses interannual variability in the climate response to large Pinatubo-sized volcanic eruptions, including the NH winter mechanisms and ENSO response (Zanchettin et al., 2021). Here, we use the volc-pinatubo-full VolMIP simulations run in GISS Model E2.1 under varying background conditions to investigate variability in the annual to interannual

climate response. In particular, we seek to answer the following questions:

- Do background ENSO and NAO conditions account for some inter-ensemble variation of the post-eruptive response?

- Do background ENSO and NAO conditions cause small variations in the climate system or uniquely different response mechanisms to volcanic eruptions?

- Does GISS model E2.1-G support proposed mechanisms of change in ENSO and the first NH winter?

## 2   Model Description and Experimental Setup

To investigate the Pinatubo response under different background conditions we run a large ensemble of simulations with background conditions sampled using two different sampling schemes. Both sets of simulations are run in accordance with VolMIP protocol with a pre-industrial atmosphere in GISS Model E2.1.

### 2.1   The Model

All model simulations are run in GISS Model E2.1 (E2-1-G in CMIP6 archive) with fully coupled ocean-atmosphere modules and in correspondence with CMIP6 protocols. GISS Model-E2.1 has a horizontal resolution of 2 degrees latitude by 2.5 degrees longitude and 40 vertical layers (which are more more densely layered close to the surface and get progressively coarser going upwards into the stratosphere). All ensembles are run with a fully dynamic mass-converting free surface Russel ocean model (Russell et al., 1995) now referred to as the GISS ocean, denoted 'G' in GISS E2.1-G. The atmosphere is represented with

non-interactive (NINT) aerosols. Thus, ozone and other aerosols are pre-determined by CMIP6- specified model inputs.

    The current CMIP6 model of E2.1-G ENSO representation has improved significantly upon E2 (CMIP5) (Schmidt et al., 2014) on correlated global changes in temperature for all ocean representations (including the GISS ocean used here). The model shows a spectral density of ENSO events peaking at a 5 year period (Kelley et al., 2020) showing a slight bias in frequency, although the relative strength of the ENSO cycle is reasonable. E2.1 also shows a higher than average standard

deviation in NAO patterns when compared to observations and other models (Orbe et al., 2020). Thus we note the model has larger variability in the NAO, likely linked to the model's increased frequency in ENSO events (Kelley et al., 2020).

### 2.1.1   Model Simulations and Sampling

After a 6000 year control run spinup, 400 years are chosen as sampled based on background ENSO and NAO conditions as ensemble years. Prior to sampling, control run simulations are run with a pre-industrial atmosphere for a total of 400 years.

Simulation years are sampled for ENSO and NAO background conditions using the VolMIP protocol for 'volc-pinatubo-full' simulations (Zanchettin et al., 2016). Specifically, from a 400-year monthly control run we sample for positive, negative and





neutral conditions of each background condition and co-condition. In total, we sample 9 years from each co-condition for a total of 81 VolMIP sampled simulations. ENSO and NAO indices of VolMIP-sampled years in comparison to the full control run are displayed in Figure S1.

Once desired years have been chosen, the simulation is begun the year before the desired condition. For example, if our desired background conditions occur in the winter of year 8001 in the control run, the ensemble simulation will begin in year 8000, with the eruption occurring in June of 8000. Thus, simulation years are chosen based on the control background conditions in the first post-eruptive winter. The anomaly (perturbed-control) of these VolMIP simulations thus show how the climate system changes from the sampled background conditions that occur in the absence of volcanic aerosols.

In addition to the VolMIP runs sampled from background conditions, we also sample 50 additional runs randomly from the same control run (henceforth referred to as Random Samples.) From these 50 randomly sampled years, 10 overlap with already sampled VolMIP simulation years. Thus, 40 additional simulations (identical to VolMIP simulations except for random sampling) were also run with NINT atmosphere and the GISS ocean for a total of 121 simulations. Background conditions in the control run are approximately randomly distributed (Figure S1). Thus, the set of randomly sampled simulations are used as
more representative sample of conditions occurring in the natural world.

     After ensemble years are selected according to the two sampling schemes (VolMIP and random samples), volcanic simulations are run in GISS-E2-1-G in accordance with VolMIP protocol (Zanchettin et al., 2016). Volcanic aerosols are prescribed based on CMIP6 Pinatubo aerosol climatologies (Thomason et al., 2018) as a function of height, latitude, and time beginning on the 6th month of the simulation (June) to emulate the Pinatubo eruption.

We examine how the climate response under volcanic conditions differs from control conditions by processing all results as anomalies from the equivalent control period (response= perturbed-control). Thus, each ensemble is analyzed as an anomaly from control conditions, effectively filtering out background conditions in the response and looking only at change in the climate signal due to volcanic aerosol forcing. This approach differs from approaches which take anomalies from a historical time period rather than from control conditions.

## 175 3 Results

### 3.1 Radiative Forcing

Prescribed volcanic sulfate aerosols in the stratosphere cause changes in shortwave and longwave radiative forcing by reflecting solar radiation and absorbing infrared radiation. These affects are measured by changes in incoming and outgoing radiation at the surface and top of atmosphere. Beginning the month of the eruption, global shortwave radiation reaching Earth's sur-
face decreases, with ensemble mean forcing peaking at -3.27 W m$^{-2}$ the December after the eruption. This magnitude of volcanic radiative flux is in good agreement with both other model studies and satellite-based observations (Schmidt et al., 2018). Changes in shortwave forcing at the top of the atmosphere shows an almost identical pattern with decreases peaking in December at -3.83 $\frac{W}{m^2}$. Ensembles also show changes in longwave radiative forcing at the top of the atmosphere with an increase in longwave forcing at the top of the atmosphere beginning on the first October after the eruption, lasting consistently





for one year and decaying after the second fall. Longwave impacts lag behind shortwave forcing, with a peak occurring the July after the eruption with a magnitude of 1.64 $\frac{W}{m^2}$. There is minimal variation in the observed radiative forcing response between ensembles, indicating that such responses are not significantly impacted by background conditions. At peak forcing the composite of 81 simulations have a standard deviation of only 0.020 $\frac{W}{m^2}$ for shortwave forcing at the surface and 0.0010 $\frac{W}{m^2}$ for longwave forcing at the top of the atmosphere. Shortwave and longwave radiative impacts for volcanic sulfate aerosols are presented in S2.

### 3.2 Surface Temperature

Global surface temperature decreases the first year after the eruption with an anomaly peaking at a -0.35°C the first spring after the eruption. Ensemble groups with different background ENSO phases show some variability in the mean global surface temperature response to volcanic eruptions. These differences between ensemble groups are, however within the ensemble variations (Figure 1). The tropical [20 °S - 20 °N ] surface temperature reduction is stronger than the global mean surface temperature response, with an ensemble average decrease of -0.43 °C. Tropical surface temperature anomalies also vary between different background conditions, particularly in the first winter. Surface air temperature in the tropics decreases by an average of 0.2 degrees mores in positive and neutral ENSO simulations than in negative ENSO simulations (Figure 1). This variability between negative ENSO groups compared to other ensembles is significant for the first 7-13 months after the eruption as verified by a pairwise comparison ANOVA test (an analysis technique which verifies the statistical variance between two groups).

Neither global nor tropical surface temperature re-equilibrate to normal conditions (anomaly of zero) before the end of the 3-year simulation due to the high thermal capacity of the ocean. This is expected as global temperature effects can take as long as a decade to return to normal (Stenchikov et al., 1998). Here we instead focus on the regional climate impacts on the inter-annual time scale. Background NAO conditions cause no large scale variations in the surface temperature response and thus are not pictured. The influence of NAO background conditions on regional scale surface temperature is further discussed in Sect. 3.4.3.

### 3.3 ENSO Response

Figure 2 shows the monthly Nino 3.4 Index (filtered to remove the seasonal signal) for positive, negative and neutral ensembles. Positive, neutral and negative ENSO ensemble groups all show negative, La Niña-like sea surface temperature anomalies in the first post-eruptive winter. Negative sea surface temperature anomalies, however, are strongest for positive and neutral ENSO conditions with mean peak decreases of -0.61 and -0.67 °C, respectively, consistent with changes in tropical surface temperature. In positive ENSO ensembles the sea surface temperature relaxes towards mean temperatures from warmer-than-average conditions. Negative ENSO ensembles show little variation between control and perturbed simulations, suggesting that a cooler-than-average tropical sea surface temperature will be affected little by volcanic perturbations. The lack of a clear ENSO response for La Niña ensembles is consistent with (Predybaylo et al., 2017) where La Niña ensembles showed no significant ENSO anomaly. Unlike results of earlier studies (Pausata et al., 2020; Khodri et al., 2017; Predybaylo et al., 2017)



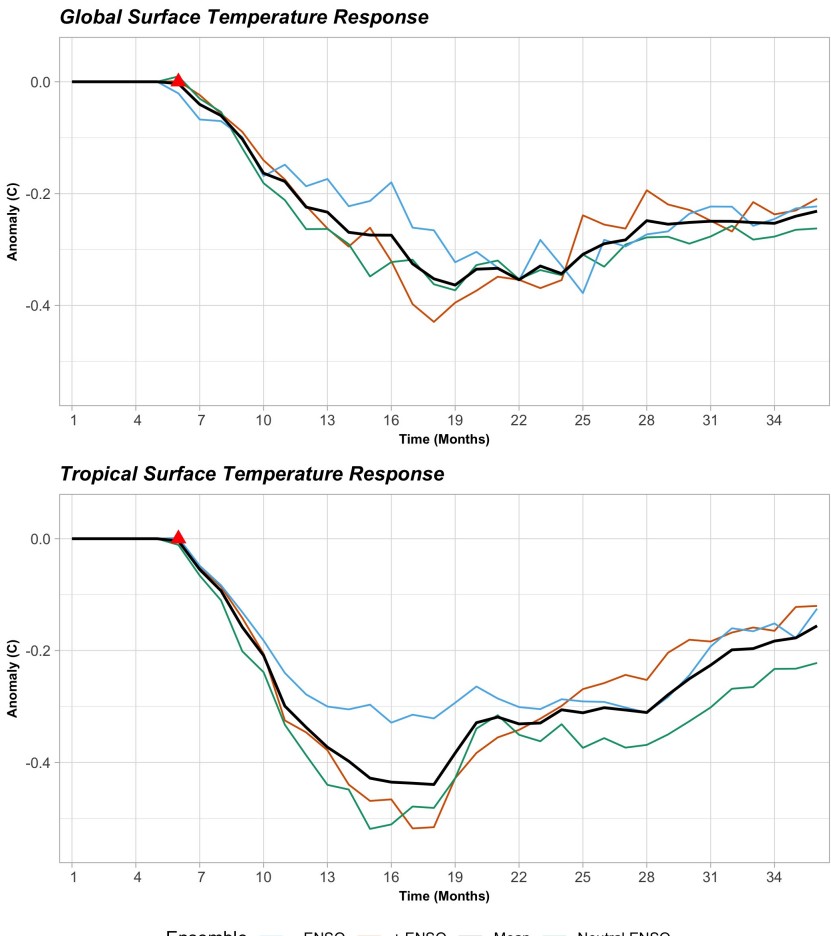

**Figure 1.** Anomalous global (top) and tropical [20°S- 20°N] (bottom) temperature response for ensembles varying by background ENSO condition. Negative ENSO simulations (blue) have weaker surface cooling, particularly in the tropics, during the first post-eruptive spring. Positive and neutral ensembles have similar surface cooling at both the global and tropical scale.

we find no El Niño anomalies in these simulations. Our findings do, however support the idea that ENSO response is dependent on pre-conditioning or background conditions in the tropical pacific (McGregor et al., 2020).

We do not differentiate here between Central Pacific and Eastern Pacific El Niño events as in Predybaylo et al. (2017). The small inter-ensemble spread for positive ENSO simulations, however, suggests that there is little difference between the two in our model representation. Overall, as all ensembles show varying degrees of cooling in the first post-eruptive winter, background ENSO condition does not greatly impact the evolution of the ENSO signal.



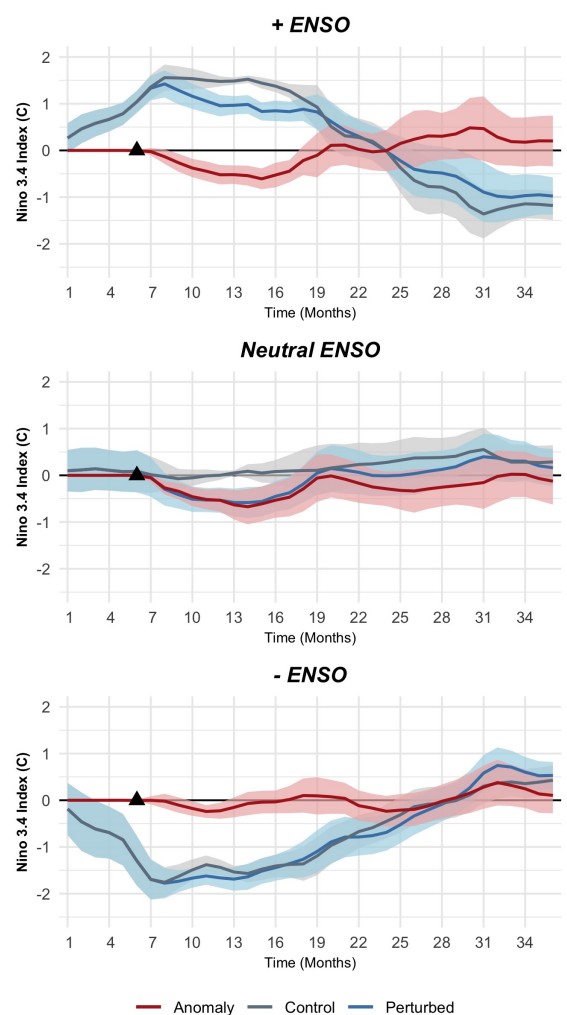

**Figure 2.** Monthly, seasonally-detrended time series of changes in the Monthly Nino 3.4 ENSO Index under different ENSO background conditions. Positive and negative ensembles show a relaxation of the index response to mean sea surface temperature conditions. Red shading shows the 95% confidence interval for the anomalous response from control conditions.

## 3.4 Northern Hemisphere Response

The tropical response depends weakly on background ENSO phase but not at all on background NAO phase. We now turn to the response in the Northern Hemisphere, which has also been widely discussed as responses also vary greatly between model studies. Here, we consider how background NAO conditions impact the Northern Hemisphere climate response, particularly in the first winter.



### 3.4.1 NAO Response

Figure 3 shows the monthly NAO index (based on 500 mb geopotential height) with the seaonal signal removed throughout the 3 year simulation period for positive, neutral and negative NAO groups. Regardless of the background phase of the NAO (positive, negative or neutral), the NAO relaxes towards mean conditions. For positive NAO ensembles, this is shown by a robust decrease in the NAO index, peaking at 69.7 mb in the February after the eruption in a winter that would normally have a high NAO index. Negative NAO ensembles, on the other hand, show a robust increase in the NAO index from control

simulations. On average, negative NAO ensembles have an increase in the NAO index by 88.5 mb peaking the first February after the eruption. Neutral NAO ensembles have no robust signal in their response, showing some increase in the NAO index during the first and second winter, but with significant ensemble spread.

These findings suggest that the prescribed volcanic aerosol forcing relaxes the extreme conditions of the NAO that are otherwise present in control runs. For negative NAO ensemble years, this causes an anomalous strengthening of the pressure

dipole between the Azores high and Icelandic low regions when eruptions occur under negative NAO conditions. The opposite is true for positive NAO simulations, where the dipole between these two pressure systems appears to weaken in comparison to control conditions the first winter after. Given this robust change in pressure in the North Atlantic during the first winter, we continue to discuss how these changes are seen through other polar dynamic pathways.

### 3.4.2 Polar Dynamic Changes

Modelled changes in the North Atlantic geopotential height dipole (quantified by the NAO index) are accompanied by other changes in zonal winds and atmospheric temperature. In particular, negative NAO ensembles exhibit strengthening westerly zonal winds at latitudes above 60 °N. Strengthening of westerly circulation is most robust high in the atmosphere, peaking at 18 m/s at the 50 mb level ($\sim$ 19 km) in the first winter. These patterns propagate down the atmosphere reaching the surface with a strength of 5 m/s.

These changes in zonal winds are accompanied changing patterns in 500 mb geopotential height ($\sim$ 5.5 km) over the polar region (60-90 °N). For negative NAO ensembles, geopotential height decreases by an ensemble average of 100 mb ($\sim$ 15.5 km) in the first winter. There are also moderate increases in geopotential height in the Atlantic basin near the Azores (20–55°N; 90 °W–60 °E).averaging around 50 mb. These changes in pressure and wind are indicative of strengthening of the polar vortex and a positive phase of the NAM. The opposite occurs for positive NAO ensembles, consistent with a decrease in the strength

of the polar vortex and a negative phase of the NAM. This pattern also suggests that the observed increase in NAO index for background negative NAO ensembles is driven primarily in pressure changes over the polar region. Neutral NAO ensembles show no clear patterns in their response.

### 3.4.3 Atmospheric Temperature

Analysis of changes in temperature in the lower stratosphere illustrate how volcanic eruptions impact atmospheric temperature

under different background conditions. Specifically, we examine temperature anomalies using Microwaved Sounding Unit



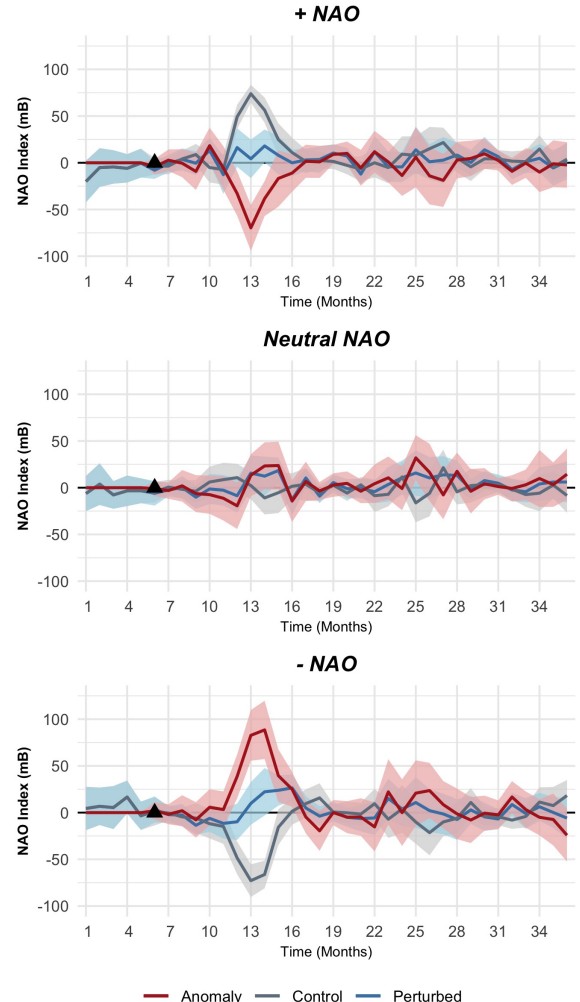

**Figure 3.** Monthly, seasonally-detrended time series of changes in the monthly NAO Index under different NAO background conditions. Red shading shows the 95% confidence interval for the anomalous response from control conditions. Positive NAO ensembles show a robust decrease in the NAO index in the first winter (t=12:14), while negative NAO ensembles show a robust increase.

temperature metrics in the lower stratosphere (MSU TLS) which are based on remotely sensed temperature data with weighted averaging based on height (Miller et al.). Figure 4 shows the anomaly in MSU temperature in the lower stratosphere for the first Boreal winter across latitude and time. All ensembles exhibit robust tropical tropospheric warming peaking at 2.5 °C, tapering off toward the south pole.

The temperature anomaly north of 60 °, however varies significantly between simulations. Positive NAO ensembles show an anomalous warming in the stratosphere reaching an average of 2.5 degrees at the north pole. Negative NAO ensembles show




stratospheric cooling anomalies reaching an average of 5.8 °C at the pole. Neutral NAO ensembles, again fall in between these extremes falling close to the mean.

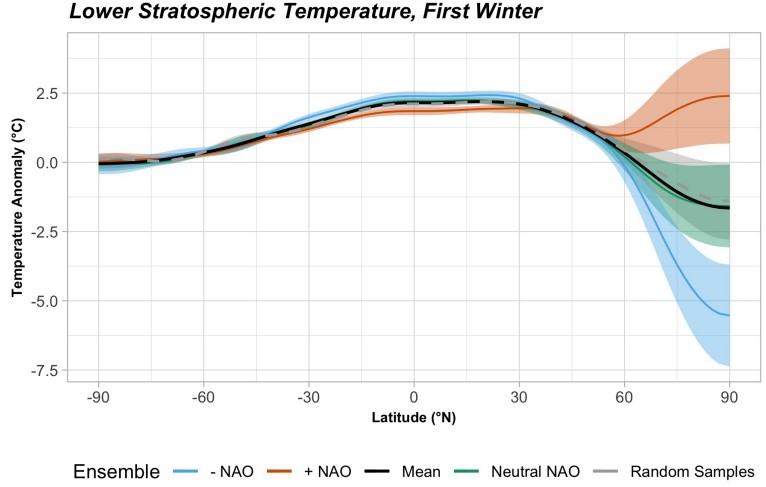

**Figure 4.** The change in lower stratospheric temperature derived from Microwave Sounding Units in the first winter (December-February) after the eruption. Shading denotes the 95 % confidence interval for each ensemble group. Tropical stratospheric warming occurs for all ensembles, however the high northern latitude response varies greatly between different background conditions. Negative NAO ensembles show cooling in the high latitude lower stratosphere while positive NAO ensembles show significant warming.

For all ensembles, volcanic forcing smooths out meridional temperature gradients in the first winter that are present in control
conditions. Thus, negative NAO ensemble simulations, which would normally have a weak meridional temperature gradient, increase the high northern latitude gradient in the first winter after the volcanic eruption. The opposite occurs for positive NAO ensembles, where higher than average temperature gradients are decreased to mean conditions (Figure S3).

Changes in the polar stratospheric temperature drive the strength of the equator-to-pole temperature difference. Negative NAO ensembles drive an increase in the equator-to-pole temperature difference driven both by warming of the equatorial lower
stratosphere and cooling of the high latitude lower stratosphere. Positive NAO ensembles, on the other hand show little change in the equator-to-pole temperature difference as both the equatorial and polar lower stratosphere experience warming. Neutral NAO ensembles fall somewhere in the middle with a moderate increase in the equator-to-pole temperature difference. To further investigate if an enhanced equator-to-pole temperature difference correlates with an increased polar vortex circulation, we use a simple regression as done by Polvani et al. (2019) with each of our 81 VolMIP ensembles. Figure 5 shows that changes
in the equator-to-pole temperature gradient in the first winter strongly correlates with increased polar vortex strength in the first winter. There is also a correlation between the observed winter warming anomaly and polar vortex strength ($R^2 = 0.40$) indicating that a strengthening of the polar vortex often corresponds with winter warming. The correlation between vortex strength and winter warming is stronger than in Polvani et al. (2019), which could suggest that larger ensembles are required to find a significant signal but does not suggest that a strengthened polar vortex alone is the cause of observed winter warming.





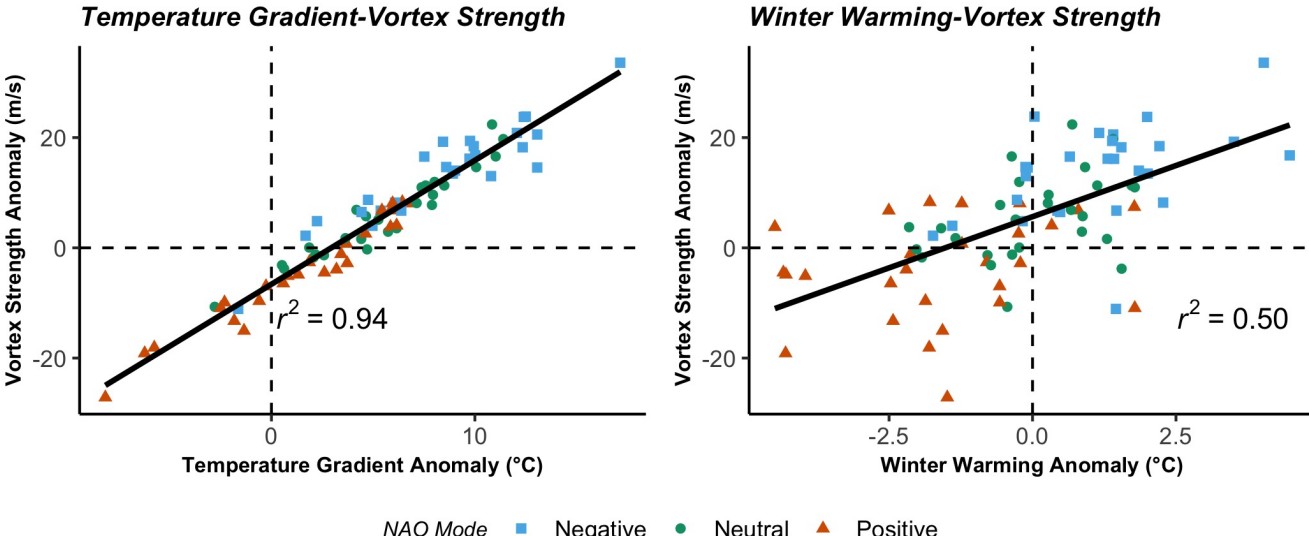

**Figure 5.** Regressions of the a) Equator-to-pole temperature gradient (temperature at 50 mb at equator-temperature at 50mb at poles) vs. stratospheric polar vortex strength ($u_{50}$ at 61 °N) and b) winter warming vs. polar vortex strength, all in the first post-eruptive winter. All 81 VolMIP ensembles are plotted with shape and color corresponding to the background NAO phase. $R^2$ values are displayed for each regression.

### 3.4.4 Winter Warming

Having discussed dynamic changes in the NH, we now discuss the strength of the winter warming response across different background conditions. Figure 6 shows the mean surface temperature anomaly in the first Boreal winter (DJF) after the eruption for positive, negative, and neutral ensemble groups. Most areas (where shading is grey) do not experience any robust difference between background NAO phases. Northern Eurasia and Greenland, however, have significantly different responses between positive and negative NAO conditions. Positive NAO ensembles experience cooling over Eurasia and warming over Greenland. Negative NAO ensembles show the opposite, with significant warming over Eurasia and cooling over Greenland. Neutral NAO ensembles show a weak warming signature similar to negative NAO ensembles. The winter warming signature (measured as a mean of 27 ensembles) is strong only for the negative NAO group.

While negative NAO ensemble group means show a significant winter warming response, we look now at variation within NAO groups. Figure 7 shows a box plot for the winter warming in Eurasia (40-70 °N, 0-150 °W) of simulations grouped by NAO phase, with all VolMIP ensembles, and for the 50 randomly sampled simulations for comparison. For comparison, we also include anomalies from historical conditions for direct comparison with other studies (Polvani et al., 2019; Driscoll et al., 2012).

With anomalies from control conditions negative NAO ensembles show mean warming over Eurasia, with few ensembles showing a negative anomaly. Positive NAO ensembles all experience a cooling temperature anomaly in Eurasia. The neutral





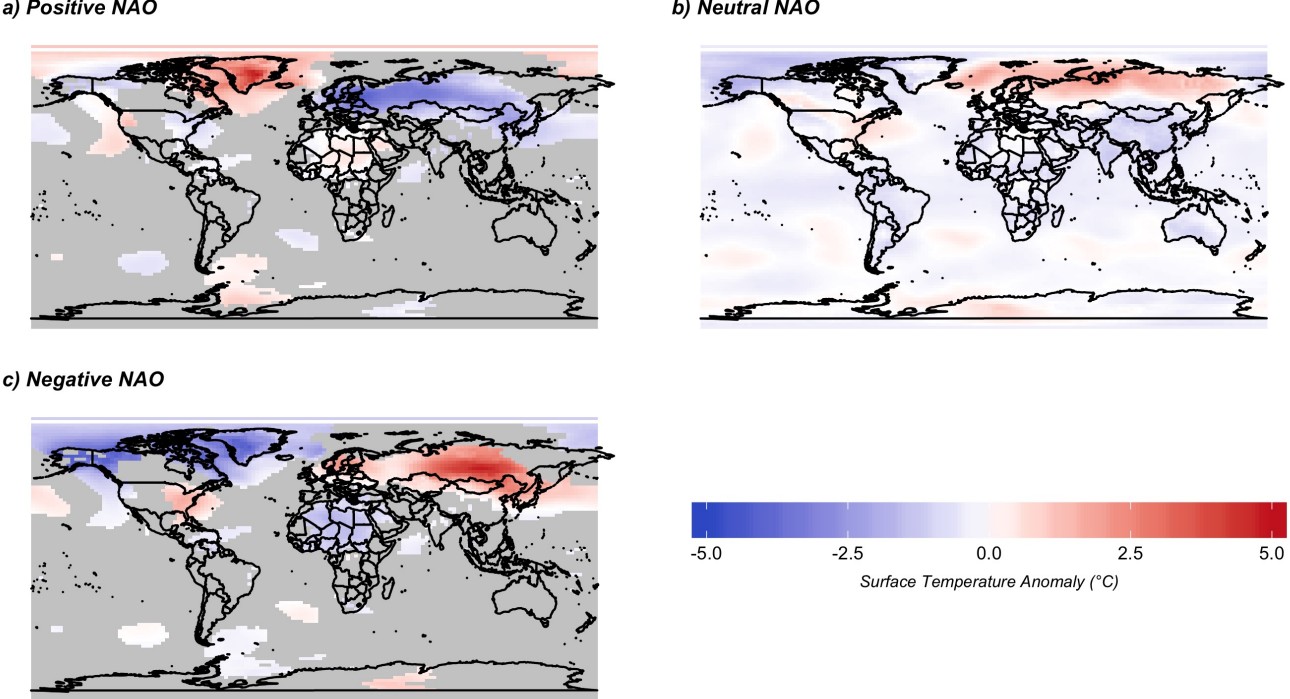

**Figure 6.** Average surface temperature anomaly in the first (DJF) winter after the eruption for positive (top left), neutral (top right) and negative (bottom left) NAO ensembles. Anomalies are taken from control conditions. Greyed areas for positive and negative ensembles denote confidence below 95% in the difference between positive and negative ensemble groups.

group of ensembles has a mean around zero degrees of warming, but is slightly skewed to a positive temperature anomaly. Plotting all VolMIP ensembles together shows a large variation in the temperature response due to including all background conditions together. The randomly sampled runs have a distribution similar to the neutral NAO ensemble groups, suggesting that extreme background conditions, such as very negative NAO or positive NAO phases, are less common in the climate

system than in our sample.

Anomalies taken from historical conditions show no significant forced response, likely because the control run anomalies are driven by changes in the NAO from control conditions. These extreme background NAO conditions are evident as the control anomalies show cooler than average conditions for negative NAO ensembles and warmer than average conditions for positive NAO ensembles. There is, however, no significant difference between the perturbed (with volcanic forcing) and control (with

no volcanic forcing) winter warming response for all ensemble members (All VolMIP and Random Samples) or for neutral NAO ensemble members.

In addition to decreasing variability in the response, ensemble groupings also impact the probability of observing warming in the model (when considering anomalies taken from control conditions). Table 1 shows the probability of simulations show-





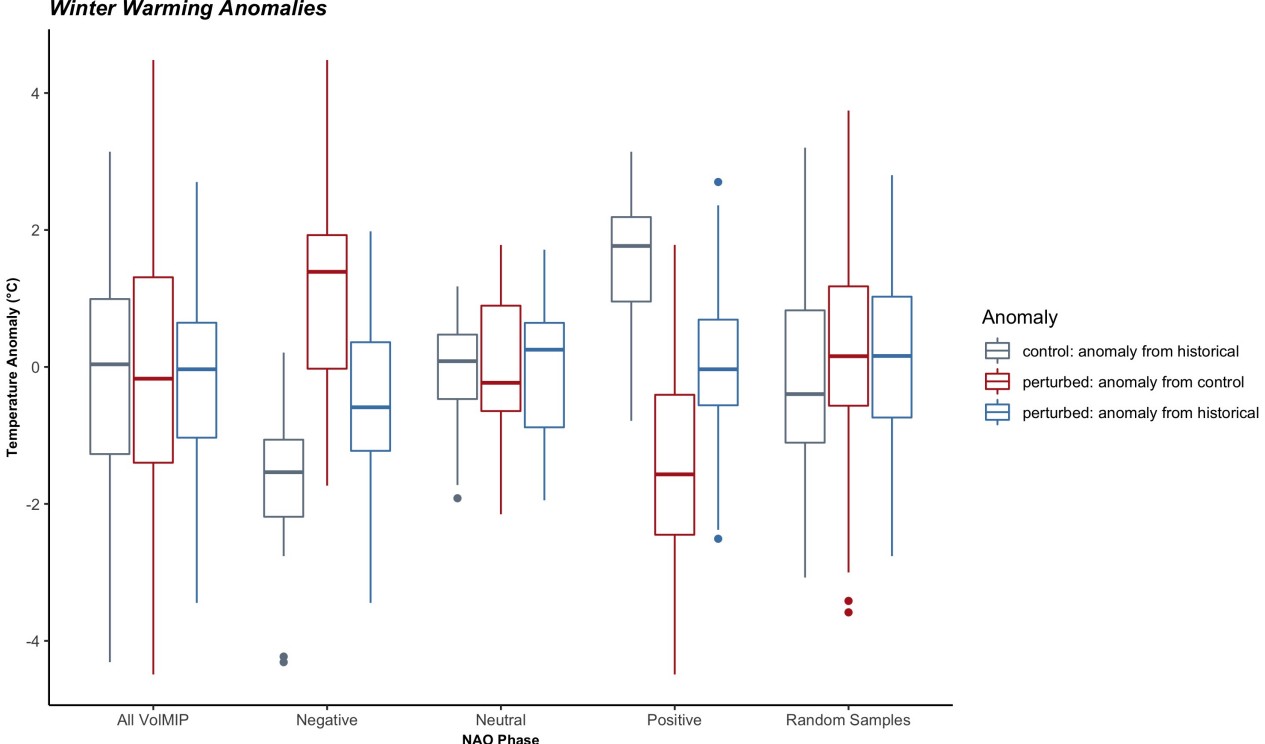

**Figure 7.** Simulated Boreal Winter Warming (December-February) response in Eurasia the first year after the eruption for both control (no volcanic forcing) and perturbed (with volcanic forcing) runs. Control anomalies (grey) are taken from the mean winter surface temperature for the five years prior to the eruption. Perturbed anomalies are shown both with anomalies from control conditions (red) and from historical conditions (blue.) Box plots are shown for all 81 simulations (All VolMIP), for each background condition group (Negative, Neutral, Positive) each with 27 simulations and for all 50 randomly sampled runs (Random Samples).

ing winter warming for varying background conditions calculated using the 81 VolMIP runs and the 27-member background
condition groupings. While not all negative NAO ensembles show a winter warming response, the probability of observing
winter warming increases greatly for negative NAO background conditions in comparison to neutral and negative conditions
when using paired anomalies. The probability for observing warming in one ensemble is low (32%). The probability of warm-
ing given negative NAO background conditions, however, is higher (60%). Thus, while these simulations still show the large
variation in surface temperature responses the first winter after the eruption, background conditions can impact how likely a
warming response is in a large group of ensembles.





**Table 1.** Winter Warming Probabilities

| Condition | Percent Probability |
| --- | --- |
| $P$(Warming) | 32 % |
| $P$(Warming \| NAO +) | 7.4 % |
| $P$(Warming \| NAO Neutral) | 22 % |
| $P$(Warming \| NAO -) | 60 % |

*Probabilities computed with VolMIP-sampled simulations

## 4  Discussion

Background ENSO conditions show a statistically significant difference in the tropical surface temperature anomaly through the first spring after the eruption. The temperature decrease is weakest for negative ENSO ensembles, which have cooler control conditions. La Niña-like cooling is strongest for ensembles with positive and neutral ENSO background condition. In general, we find no signature of an enhanced El Niño-like anomaly, as has been suggested from other studies (Pausata et al., 2020; Khodri et al., 2017; Predybaylo et al., 2017), in any of our ensembles for the first three years after the eruption. Rather, in the GISS model we find that anomalous sea surface temperature cooling (La Niña like conditions) occurs regardless of background conditions.

The response of the Northern Hemisphere varies significantly between ensembles with different background NAO conditions both in sign and strength of responses. Winter warming anomalies occur with increased probability for ensembles with negative background NAO conditions with 60% of ensembles showing a warming response in the first winter. This warming response corresponds with an anomalous decrease in polar lower stratospheric temperature in the first winter for negative NAO ensembles, causing an increased temperature gradient between the equator and poles. A simple regression shows that positive temperature gradient anomalies are correlated with an increased strength of the stratospheric polar vortex. Negative NAO ensembles show decreased geopotential heights and increased westerly zonal wind circulation that are consistent with this strengthening polar vortex anomaly. There is a weak correlation between strengthening of the stratospheric polar vortex and the winter warming response in the first winter. Most, but not all negative NAO ensembles experience winter warming as well as a strengthening of the polar vortex, although this correlation does not suggest the lack of other response pathways. In general, positive NAO ensembles show the opposite anomalous patterns from control conditions and neutral ensembles show some weak warming and vortex strengthening anomalies. Thus while a negative NAO phase does not guarantee winter warming resulting from the equator to pole temperature difference, it does highly increase the probability winter warming will occur. These polar dynamic changes also coincide with a smoothing of meridional temperature gradients from control conditions. This response could be model-dependent, or a result of the specific way that the Pinatubo forcing is prescribed in the simulation with non-interactive aerosols.

For all simulations, the monthly NAO index in the first winter relaxes towards mean conditions. This means that for both positive and negative NAO ensembles, there is a sudden anomalous change in pressure in the North Atlantic after the eruption.





Thus, the anomalous strengthening of the polar vortex from control conditions could be due to the sudden relaxation of the NAO anomaly in the first winter. The strengthening of the stratospheric polar vortex resulting in winter warming thus only occurs when the model would have otherwise experienced weak vortex circulation in the absence of volcanic forcing. The anomalous

response is significantly impacted by these extreme background conditions that were sampled from our control conditions. We also compare our 27-member background NAO ensemble groups to a 50-member randomly sampled ensemble group. The randomly sampled ensemble group shows anomalies most similar to the neutral NAO ensemble group, suggesting that strong anomalies due to extreme NAO background conditions are less common in a representative sample. While extremely negative phases are most likely to experience winter warming such extremes are less common in the real world, possibly explaining why

warming is only sometimes observed in model simulation ensembles. These extremes in background conditions can contribute significantly to ensemble variation, particularly with a small amount of ensemble members or when ensembles are sampled with a bias in background conditions.

We also compare these anomalies from control conditions with historical anomalies as used in other studies (Polvani et al., 2019; Driscoll et al., 2012). These historical based anomalies, which take reference from mean climate conditions show no

statistically significant forced response for our ensemble members. The difference in responses between anomalies demonstrates how the choice of anomalies can significantly impact the modelled response. When analyzing modelling results using paired anomalies, background climate conditions can significantly influence strength of a given response. For example when analyzing the winter warming response under varying NAO conditions, ensembles which in the control run experienced a strongly negative NAO condition relaxed towards the mean under perturbed volcanic aerosol runs. Thus, the strength of the

winter warming response is biased due to lower mean air temperatures under the control. When using historical anomalies, we see no significant warming response for the same perturbed runs, as air temperatures are typical of historical-mean climate state (neutral NAO conditions). The difference between the modeled responses under paired and historical anomalies was also highlighted by Zanchettin et al. (2021), where the choice of anomaly was shown to impact some ensemble mean responses, but can mitigate the effect of sampling biases.

Simulations have been constrained to examine the climate response with a protocol that eliminates some sources of variability. In particular, VolMIP compliant simulations used here are run with NINT aerosols and represent pre-industrial conditions. Thus, they cannot be directly compared to Pinatubo simulations which have industrial greenhouse gases in the atmosphere. These runs also do not account for changes in ozone concentration observed after eruptions which may also influence changes in stratospheric circulation (Stenchikov et al., 2002). The NINT atmospheric representation also dictates that aerosols evolve

exactly as perscribed, making aerosols insensitive to states of the stratosphere and troposphere. Other runs with interactive aerosols are necessary to understand if the dynamics of these responses are dependent on the specific perscription of volcanic aerosols.

Further, the current GISS Model E2.1 does not have a realistic representation of some key atmospheric components such as the Quasi Biennial Oscillation (QBO) (Rind et al., 2020) that could also play a role in the observed circulation responses

(Stenchikov et al., 2004). Changes in the QBO could also influence the strength of the polar vortex circulation as easterly phases (such as those during the Pinatubo eruption) are likely to cause a decrease in the stratospheric polar circulation (Holton





and Tan, 1980). Here we have used the GISS E2.1-G CMIP6 compliant runs, however in a future study, we hope to examine this response in the new GISS model E2.2 which has a better representation of the QBO.

Overall, we find that background ENSO conditions have a small effect on surface temperature and ENSO response as a
cooling, La Niña like, anomaly in the tropical pacific occurs for each ensemble. The background state of the NAO, on the other hand, varies the anomalous response by relaxing background conditions in the first winter to a neutral NAO phase. If a volcanic eruption occurs during a normally negative NAO phase, these changes in turn increase the probability of observing an winter warming response from control conditions in the first post-eruptive winter. For extremes in background NAO conditions, changes in the northern hemisphere are the most robust. While often these extremes are uncommon, they likely contribute
to inter-ensemble variation and thus uncertainty in predicting the climate's response to volcanic eruptions. When the forced winter warming is defined as the average of a large number of ensembles (including all background conditions), however, the response is insignificant (mean zero). The prevalence and strength of this anomaly is influenced both by extremes in background conditions, and how anomalies are taken (either from control or historical periods.)

## 5   Conclusions

The climate response to large, Pinatubo-type volcanic eruptions is variable between models, and has here been discussed in GISS Model E2.1-G. We focus on two responses which have been studied both with observational and modelling studies: the ENSO response and Northern Hemisphere response in the first winter. 121 ensembles were run in the GISS E2.1-G model to examine how background ENSO and NAO conditions impact the modelled climate response. Our experimental setup uses a pre-industrial model with prescribed aerosols, and took anomalies from an equivalent control period run rather than a historical
climate period, allowing us to filter out background climate variability and look only at the response due to volcanic sulfate aerosols. We find that ensembles with different background NAO conditions have significantly different anomalous climate responses in the first NH winter. In particular, years which would be in positive or negative NAO conditions are relaxed to mean NAO conditions under volcanic forcing. This creates an anomalous negative and positive winter warming response for positive and negative NAO ensembles, respectively. Ensembles with different background ENSO conditions, however, show
similar anomalies between different background phases. Thus, inter-ensemble variation caused by background conditions is significant particularly when looking at the first NH winter response.

*Data availability.* All standard data from the pre-industrial control (piControl) simulations discussed here are publicly available in the CMIP6 archive through multiple nodes of the Earth System Grid Federation. Corresponding volcanic simulations used in this paper were submitted to the Volcanic Model Intercomparison Project of CMIP6 under the GISS 'volc-pinatubo-full' experiment submission and will
become publicly available as part of the CMIP6 archive by the time of this article's publication.



*Author contributions.* Conceptualization and methodology were done by AL and KT; model runs were performed by KT; Software was developed by AL, KT, and HW; Analysis and visualization of model runs was performed by HW; HW wrote the manuscript draft; AL, HW, and KT reviewed and edited the manuscript.

*Competing interests.* The authors declare that they have no conflict of interest.

*Acknowledgements.* Resources supporting this work were provided by the NASA High-End Computing (HEC) Program through the NASA Center for Climate Simulation (NCCS) at Goddard Space Flight Center. All thank NASA GISS for institutional support. Thank you to the National Science Foundation who funded this research under REU Grant number OCE 17-57602 as part of the 2019 Lamont-Doherty Earth Observatory Summer REU program. The authors would also like to thank Clara Orbe and Lorenzo Polvani for their help in analyzing results and contributing key insights from these simulations.





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
