# Peer review of "The Impact of ENSO and NAO Initial Conditions and Anomalies on the Modeled Response to Pinatubo-Sized Volcanic Forcing"

_Atmospheric Chemistry and Physics, 2023_

## Community Comment (CC1)

[revised manuscript text omitted]

---

## Author Comment (AC1)

**Dear Editors and Reviewers,**

We thank all three community and referee commenters for their careful review and consideration of our manuscript. During the revision period we have implemented significant changes to the framing and discussion of the paper to better contextualize the study with the recent analysis of the VolMIP community paper and previous studies that have analyzed the impact of initial climate states on modeled volcanic forcing. We have also significantly clarified terminology and simplified the presented results to focus on how the sampling initial conditions and anomalies impact the modeled response to Pinatubo-sized eruptions in the tropics and North Atlantic.

We have also added some new analysis to better communicate GISS-specific correlations in initial conditions, and how the analyzed response is impacted by the choice of anomaly. When possible, we also added discussion about how our results could lead to further analysis of the dynamics of the response, but do not include a full analysis of dynamic mechanisms as it is out of the scope of this paper.

It is our hope that with these revisions, the manuscript gives more clear insights to how background conditions and choice of anomalies can impact model results. We also hope to further clarify the GISS-specific elements of these responses to contextualize these findings for better use in future analyses with other models. Finally, all minor revisions will be addressed to polish the manuscript to be free of typos and errors.

As a post-baccalaureate researcher no longer funded to work on this project, I hope that these revisions have produced a manuscript that is up to the high quality standards of ACP and will provide a valuable contribution to the scientific community and look forward to further feedback in the review process.

**Sincerely,**

**Helen Weierbach on behalf of all co-authors**

**CC: Alan Robock.**

We thank Dr. Alan Robock for his review of the manuscript. Below are our responses to his comments. We believe that this version of the manuscript will present better context for our study and clarify to the reader that our study does indeed follow the sampling protocol of the Volcanic Model Intercomparison Project.

I recommend rejection or major revisions. The paper purports to be a set of calculations that follows the VolMIP protocol for volc-pinatubo-full simulations (Zanchettin et al., 2016). But unfortunately, they were not done correctly. The paper kept mentioning "background conditions" for the experiments, but did this mean the conditions at the time of the eruption? Both ENSO and NAO actually evolve in response to eruptions, so they are not background. I kept thinking, shouldn't it be "initial conditions?" Then I figured out the problem. Zanchettin et al. (2016), of which I am a co-author, says:

"Initialization is based on equally distributed predefined states of ENSO (cold/neutral/warm states) and of the North Atlantic Oscillation (NAO, negative/neutral/positive states). … The recommended ENSO index is the NH winter (DJF, with January as reference for the year) Nino3.4 sea-surface temperature index, defined as the spatially averaged, winter-average sea-surface temperature over the region bounded by 120–170° W and 5° S–5° N. The recommended NAO index is calculated based on the latitude–longitude two-box method by Stephenson et al. (2006) applied on Z500 data, i.e., as the pressure difference between spatial averages over (20–55° N; 90° W–60° E) and (55–90° N; 90° W–60° E)."

However in this paper, the ENSO and NAO states were chosen for the year AFTER the eruptions, not for the initial conditions. This is a completely different experiment, and the authors do not explain why they did it that way.

In the manuscript, we use the term "background conditions" to refer to the state of ENSO and NAO occurring in the model at the time of peak volcanic forcing. The new version of the manuscript replaces the terminology "background conditions" with "initial conditions" at the time of the volcanic eruption as defined in Zanchettin et al. 2016 and discussed in the recent VolMIP community paper.

Most importantly, this study did follow the sampling protocol for VolMIP simulations cited by Dr. Robock. Zanchettin et al. 2016 states that "*The sampled years refer to the second integration year of the VolMIP experiment, when the volcanic forcing is generally strongest. Therefore, if, for instance, year Y of the control integration matches the desired conditions for the sampling, then the corresponding VolMIP simulation should start with restart data from year Y-1 of the control, for the day of the year specified for the experiment.*"

Thus, we do sample for states of ENSO and NAO at the time of peak volcanic forcing. The manuscript does not refer to the details of this sampling but instead refers the reader to details from Zanchettin et al. 2016 stating "Simulation years are sampled for ENSO and NAO

background conditions using the VolMIP protocol for 'volc-pinatubo-full' simulations (Zanchettin et al., 2016)". [lines 150-151]. The specifics of this sampling and the terminology for sampled ENSO and NAO states are now clarified in section 2.2: Model Simulations and Sampling. The reader is also now referred to Figure 1 (previously Figure S1) which shows the sampled states of the model at the time of peak forcing in reference to all background climate states in the model, and to the figure of GISS sampled states at the time of peak forcing in comparison to other climate models for the VolMIP experiments (Zanchettin et al. 2022).

It also seems that the GISS model does not allow radiative heating of the stratosphere, which would change stratospheric circulation and affect the AO and NAO. The model also cannot produce the observed El Niño after the 1991 Pinatubo eruption.

For these reasons, the results and the conclusions in this paper cannot be supported.

Any revision would have to address the concerns below and also each of the 48 comments in the attached annotated manuscript. (Comments were stopped after line 270 because of the erroneous results.

We also acknowledge that GISS Model E2.1-G used here does allow radiative heating of the stratosphere, affecting the AO and NAO as noted by Dr. Robock. While the model does not consistently produce the El Nino signal that was observed after the 1991 Pinatubo eruption, the simulations here are not historical simulations (do not include post-industrial aerosols or initial conditions) and thus do not aim to be directly comparable to the observed volcanic response to a Mt.Pinatubo eruption.

This paper left out at least five important references, which give conflicting results to the conclusions here.

We also thank Dr. Robock for the additional important references which were not included in this preprint. These citations were not intentionally excluded, and where applicable we have added these citations to ensure proper contextualization of our findings. (See sections 1.2 and 1.3)

Lines 63-66:  You left out Zambri and Robock (2019), who showed (their Fig. 15) that no matter what the initial ENSO state, the WACCM model, in response to the 1783 Laki eruption, shows an increase SST in the Niño3.4 region of 0.5-1.0°C.

Lines 70-76:  You left out Coupe et al. (2021), who showed that cooling of the Maritime Continent and tropical Africa produced an El Niño response when forced with soot aerosols in the stratosphere.

Both papers are now included as citations in Section 1.2 ENSO Response with reference to the fact that they look at different forcing (Nuclear aerosols, Laki eruption rather than Mt.Pinatubo).

Line 88:  You left out three important papers.

Zambri and Robock (2016) shows that if you look at just the first winter after large eruptions since 1850 in the Coupled Model Intercomparison Project 5 historical simulations, most models do produce a winter warming signal, with warmer temperatures over NH continents and a stronger polar vortex in the lower stratosphere.  Zambri et al. (2017), which was written by one of the authors of the paper being reviewed here, showed the same thing for the last millennium.

Coupe and Robock (2021) showed that when there is an El Niño in the winter after a large volcanic eruption, as there was in observations after the 1982 El Chichón and 1991 Pinatubo eruptions, the NCAR CAM5 AMIP Large Ensemble shows winter warming for every ensemble member (their Fig. 1).

We have also significantly section 1.3 to both include these references and discuss evidence which both support and oppose post-eruptive winter warming.

Lines 151-152:  I don't understand how specific years were sampled for ENSO and NAO conditions.  Each of these has time scales that span different years and are usually stronger in NH winter.  So how were the years identified with respect to the phase of each of these phenomena?  Is there attention paid to the phase being strong at the time of the simulated eruption?

We thank Dr. Robock for pointing out the confusion in this section for sampling initial conditions of ENSO and NAO. The methods section has been significantly restructured in response to his and other reviewers comments and now is structured to emphasize that our 81 ensembles of VolMIP simulations were sampled with the same methodology described in Zanchettin et al. 2016 as shown in the Zanchettin el al. 2022 community paper. The methods now seek to emphasize that the only difference in sampling was done for sampling 50 'random' simulation years with no precondition on initial ENSO and NAO conditions.

Lines 170-174:  The technique of comparing simulations that start with identical initial conditions, but with and without volcanic eruptions, will not give results that identify the effects of volcanic eruptions unless multiple ensemble members are used for each experiment, because natural weather variability (chaos) will also be a large part of the differences.  Yes, the weather will be the same for a few days, but will evolve differently, so how can you determine which is causing the differences in the pairs, forcing or internal variability?

Section 2.3 now includes a more robust discussion about both paired and climatological anomalies, and how these two choices anomalies vary in how they display other non-volcanic forcing and background climate conditions. We hope that this section gives readers some clarity on why different anomalies may show different results.

I don't understand the NAO results at all. Did your implementation of volcanic aerosols allow them to heat the stratosphere in the Tropics? If not, you did not force the climate system correctly. This tropical heating should produce a positive NAO, which should produce winter warming because of the increased polar vortex. See Coupe and Robock (2021).

References to be cited

Coupe, Joshua, and Alan **Robock**, 2021: The influence of stratospheric soot and sulfate aerosols on the Northern Hemisphere wintertime atmospheric circulation. *J. Geophys. Res. Atmos.*, 126, e2020JD034513, doi:10.1029/2020JD034513.

Coupe, Joshua, Samantha Stevenson, Nicole S. Lovenduski, Tyler Rohr, Cheryl S. Harrison, Alan **Robock**, Holly Olivarez, Charles G. Bardeen, and Owen B. Toon, 2021: Nuclear Niño response observed in simulations of nuclear war scenarios. *Communications Earth & Environment*, 2, 18, doi:10.1038/s43247-020-00088-1.

Zambri, Brian, and Alan **Robock**, 2016: Winter warming and summer monsoon reduction after volcanic eruptions in Coupled Model Intercomparison Project 5 (CMIP5) simulations. *Geophys. Res. Lett.*, 43, 10,920-10,928, doi:10.1002/2016GL070460.

Zambri, Brian, Allegra N. LeGrande, Alan **Robock**, and Joanna Slawinska, 2017: Northern Hemisphere winter warming and summer monsoon reduction after volcanic eruptions over the last millennium. *J. Geophys. Res. Atmos.*, 122, 7971-7989, doi:10.1002/2017JD026728.

Zambri, Brian, Alan **Robock**, Michael J. Mills, and Anja Schmidt, 2019: Modeling the 1783–1784 Laki eruption in Iceland, Part II: Climate impacts. *J. Geophys. Res. Atmos.*, 124, 6770-6790, doi:10.1029/2018JD029554.

Review by Alan **Robock**

**RC1: Davide Zanchettin:**

We thank Dr. Davide Zanchettin for his thorough and helpful response to the manuscript. His comments were particularly helpful for understanding how to clarify methodology and analysis to be consistent with other studies and improve readability, identify new analysis needs to understand more dynamics of the response and contextualize the GISS findings with the multi-model VolMIP analysis.

In response to his comments we undertook several specific manuscript edits and further analysis for better contextualization. Specific responses to his comments are included below.

I read with interest the manuscript by Weierbach et al. and I think it could be a valuable contribution to the VolMIP special issue, but pending revisions as detailed below. In my opinion, the revision should account for improvements in methodology as well as presentation and writing.

The main element of novelty of this study is the found "dampening" of NAO anomalies by the volcanic perturbation and its consequences for the post-eruption winter warming. However, this novel aspect is not fully investigated/understood, while there are results that seem ancillary. Since this single model study builds on the multi-model experiment volc-pinatubo-full, I think it should more strongly connect to the descriptive paper of the experiment (Zanchettin et al., 2022) for the general description to then focus on the novel aspects and possible model specificities that characterize GISS-E2.1-G. So, I recommend digging more into the main result and reduce the presentation of results that seem less insightful for the focus of the study (for instance paragraphs 3.1 and 3.2).

The new version of the manuscript focuses on how the dampening of NAO anomalies are impacted both by sampling (VolMIP sampling vs random sampling) and by the choice of anomaly (paired anomaly vs. climatological anomaly). The re-worked methods section focuses on emphasizing the importance of these two aspects, and contextualizes the work more with the recent Zanchettin et al. 2022 paper. We have also moved results that are less insightful to the supplemental materials such as the original sections 3.1 and 3.2.

An obvious question that raises but remain unanswered is the cause of the different stratospheric response at high-latitudes in the different NAO sub-ensembles. There is no real investigation or discussion about the possible underlying mechanisms, but this seems to me central to establish a dependency on initial conditions. Also, this limits to put the GISS results in the context of future analyses with other models contributing to the same experiment. At least, I suggest checking the literature, for instance the Toohey et al. paper, for insights.

To better understand possible mechanisms we have further contextualized the findings of previous studies such as Toohey et al. in section 1.3 where we now discuss in more details what

evidence there is in previous modeling studies to both support and oppose proposed winter warming and corresponding strengthening of the polar vortex in the context of initial conditions.

Also, the mentioned strong correlation between ENSO and NAO in GISS could be relevant in shaping post-eruption NAO variability but is not considered. This could "bias" the results, or render them model specific, so should be at least discussed. Having a large ensemble, this could be checked by further stratifying responses around both, NAO and ENSO states.

While we have not expanded the presented results with underlying mechanisms of the different responses based on ENSO and NAO conditions, we have expanded to discussion of biases between NAO and ENSO within the GISS model. Section 2.2 now includes the figure which shows the correlations between the ENSO and NAO indices specified by the VolMIP experiments for all 400 years of the control period, and for the sampled years used in this study.

The "propagation" of anomalies from the stratosphere to the troposphere is not shown but should. This seems to me a necessary step to establish that it is the stratosphere that drives the tropospheric changes under all conditions. This could be done, for instance, with a figure of vertically resolved zonal average zonal wind anomalies at different time steps, for the different initial conditions. Another valuable figure could include maps of gridded 500 hPa geopotential height anomalies linked with the different initial conditions. Such maps would illustrate possible asymmetries and specificities in both, the meridional structure of atmospheric circulation and across the sub-ensembles with different initial conditions (as mentioned above, clarify potential remote sources of anomalies over the North Atlantic).

This paper aims to discuss what ENSO and Northern Hemisphere responses exist in ensembles with GISS Model E2.1-G. While examining propagation of anomalies from the stratosphere to the troposphere would give further insight into the proposed mechanisms it is not included in this paper because it is out of scope of this work. To better reflect this scope, we have simplified the text in the introduction where specific mechanisms of strengthening of the stratospheric polar vortex was previously discussed, to focus on the statistical impacts of sampling and anomalies in a way that compliments the recent Zanchettin et al. 2022 community paper.

As a last note on the NAO response, in their admittedly simpler (full-ensemble) analysis, Zanchettin et al. (2022) reported a tendency "toward positive NAO anomalies in the first post-eruption winter in GISS-E2.1-G". An explicit discussion here about this result seems appropriate.

Because the NAO response was only presented for the NAO +/0/- groups, this finding as pointed out by Zanchettin et al. was not adequately discussed. We have performed the additional analysis which shows the equivalent NAO anomalies for the full 81-member VolMIP sampled ensemble, now in the four panel plot shown below. The new version of the manuscript now includes reference to this finding from Zanchettin et al. 2022, and discusses how the tendency toward positive NAO anomalies varies between different ensembles (see figure below for illustration).

[Figure]

For consistency we have also shown the equivalent full 81-member mean response of ENSO in our updated version of Figure 2, where the addition of the 'All VolMIP Ensembles' highlights that despite a significant spread in control and perturbed conditions of the Nino 3.4 region, there is a consistent cooling of the sea surface temperatures in this region following the volcanic eruption for all ensembles.

[Figure]

Further regarding analyses, as a general comment, statistical support in the assessment of differences across ensembles (or sub-ensembles) is crucial. The authors mention ANOVA at some point, but do not show the associated p-value for the significance. Significance is then reported or only mentioned only occasionally. This must be amended. Also, all figures should report the ensemble envelope, not just the mean, as this alone can be deceptive. I recommend including a section 2.2 on "data analysis" where statistical methods are described, and terminology presented (see below).

Concerning the analysis of ENSO, the fact that the authors do not identify an El Nino-like response is very likely linked to the fact that the Nino3.4 index "as is" includes the volcanically induced cooling of the whole tropics, which must therefore be removed before investigating dynamical responses of ENSO. The most used approach is based on "relative SST" and is discussed in several papers, for instance Khodri et al. (2017) and Zanchettin et al. (2022). I strongly recommend the authors to revise the ENSO analysis to account for this. Note that using the relative SST method, Zanchettin et al. (2022) report the GISS-E2.1-G "showing a slight warm ENSO anomaly in 1992 in the ensemble-mean", so contrasting the result reported here in this version of the manuscript.

A new section is now included in methods (Section 2.3– Data Analysis and Anomalies) that seeks to further a) more robustly define the differences in paired vs. climatological anomalies and b) define the statistical analysis done for the presented results. Results do include confidence intervals where ensemble means are presented, but do not show confidence intervals for results where either a) each ensemble is presented as an individual sample or b) ensembles are presented as box and whisper plots where rather than confidence intervals statistical quantiles are presented.

We have additionally added ANOVA test results for sections where the difference between initial condition ensemble groups is a main point of the results in the winter warming analysis. For example, for the DJF Winter Warming response between different NAO initial condition groups, we present the results of the ANOVA Test in section 3.2.4 as:

" An ANOVA test shows there is a statistically significant value between paired anomaly ensemble groups with different initial NAO conditions in the VolMIP ensembles with a F-statistic value of 22.78 and a p-value  of 1.62e-08."

Then, I read the comment by Alan Robock, and I agree that the phrasing may lead to some misunderstanding especially for those not familiar with the VolMIP protocol. This seems to me highlighted in this manuscript as the wording is often vague and does not stick to a well defined terminology.

The issue: For the volc-pinatubo experiments the selection of initial conditions was based on the idea to sample initial conditions that would lead, without the eruption, to a "controlled" diversity of ENSO and NAO conditions, ultimately to avoid sampling biases of internal variability. **Therefore, the sampled ENSO and NAO refers, in time, to the first post-eruption winter as correctly described in this manuscript.** What is sampled, therefore, are initial states that may capture preconditions to (or developing) different states of ENSO and NAO. This issue is presented in the volc-pinatubo-full multi-model ensemble by Zanchettin et al. (2022), where the choice, by some groups, to target the last per-eruption winter rather than the first post-eruption winter is also illustrated (see section 2.2.8 and Figure 2 of Zanchettin et al., 2022).

The same paper also discusses an adjustment of the VolMIP protocol for future experiments so that "the ENSO mean state and tendency on the period from the last pre-eruption winter to the

onset of the eruption is considered instead of the state during the first post-eruption winter as in the original VolMIP protocol." So, indeed this acknowledges a potential issue that concerns the VolMIP protocol, not this specific study.

The solution I suggest: To avoid misunderstanding I would suggest using a stricter wording with clear terminology and clear definitions. I still think it is viable to use a nomenclature for the ensembles as NAO positive/NAO negative/NAO neutral, especially if in reference to Zanchettin et al. (2022): indeed, in this paper the sub-ensembles with different initial conditions are labeled as, for instance, NAO+, NAO- and NAO0. Then, the text should be simplified as much as possible by referring generally to initial conditions, rather than specific examples that could be deceptive. For instance, at line 57, instead of "i.e. what state of NAO the climate system would normally be in" (normal refers to some average…) one should write something along these lines to be accurate: "i.e., what state of NAO the climate system would be in if the eruption did not occur" or, better, simply avoid the quoted sentence. Or, at line 126/127, one could rephrase: "background ENSO and NAO" simply with "initial conditions".

Part of the problem above is that the manuscript does not seem as polished as it should. I spotted several typos, only some of which are reported below. So, please carefully check the manuscript in the revision.

We again thank Dr. Davide Zanchettin for his thorough review of our manuscript. We have done our best to clarify the manuscript both by including a more explicit definition of anomalies and sampling protocols and by polishing the manuscript through more thorough review.

Minor/specific comments:

Line 45-50: the cited paper mainly focused on decadal changes. Another relevant paper here, focusing on ENSO and interannual time scales, is Pausata et al. (2020), already cited in other parts of the manuscript.

Lines 45-50 were rewritten to acknowledge both Zanchettin et al. 2013 and Pausata et al. 2020 in reference to both decadal and interannual variability in the climate response under different initial climate states.

Line 56: only atmospheric? This should concern all aspects of the coupled Earth system, not only the atmosphere.

I was not able to find the line referenced here, but have attempted to add reference to aspects of the coupled Earth system rather than simply to the atmosphere in all relevant areas of the introduction.

Line 57: maybe it could be useful to briefly introduce this feature of post-eruption climate evolution

Lines 56-57 have been removed and replaced with text that better defines initial climate conditions of the Earth system, rather than referring to only initial atmospheric conditions. We have also added some explanation of post-eruption climate evolution.

Line 66-69: this also concerns dynamics so I would put this in the next paragraph. I think this section is a bit a back and forth between characterization of post-eruption ENSO anomalies and mechanisms, which can be confusing for a reader. I suggest some reorganization.

The introduction, and in particular sections 1.2 and 1.3 which discuss the ENSO and Northern Hemispheric responses have been almost entirely rewritten to include important references and separate discussion between dynamics and responses.

Line 88: large numbers of ensembles à large ensembles

This has been corrected.

Paragraph 1.1.2: The paper by Toohey et al. (2014) could be cited here, as they question volcanic aerosol heating as a dominant mechanism for the post-eruption strengthening of the polar vortex.

Added reference to Toohey et al. paper and the corresponding importance of aerosol forcing and aerosol heating in the strengthening of the polar vortex.

Line 110: typo (withing)

Typo has been corrected.

Line 126-129: I must admit that I struggle to understand the exact meaning of these questions. I recommend some rephrasing. For instance, everyone would agree that different initial conditions would cause some ensemble spread (inter?) in the response even to a super eruption. So, what do you mean here? Similarly, what "small variations in the climate system" means is unclear, although this sounds like a repetition of the first question. Please also report always "post-eruption" of "volcanically forced" for the sake of clarity (for instance "post-eruption change in ENSO")

Main points at the end of the introduction have been updated to better reflect the main points of the manuscript. Here and in other places in the manuscript "post-eruptive" has been replaced with "post-eruption" and "volcanic

Line 136-138: Isn't GISS-Model E2.1 a coupled climate model? It seems you address this as only the atmospheric component. Please clarify.

GISS E2.1 is a coupled climate model. The language has been clarified and now reads "All model simulations are run in GISS Model E2.1 (E2-1-G in CMIP6 archive): a climate model with fully coupled ocean-atmosphere dynamics and in correspondence with CMIP6 protocols."

Line 148-154: I don't understand these sentences and the described method. For the VolMIP protocol, combinations of ENSO and NAO states should be sampled from a control run, then the associated states should simply be used to initialize simulations including volcanic forcing. The method that is described is confusing. Also, what is a background condition and co-condition?

The terminology in methods has been reworded for clarity. The manuscript now only defines "initial conditions" and defines these conditions in section 2.1. Initial conditions and initial co-conditions (combined states of ENSO and NAO for a given year) are also explicitly defined and illustrated in Figure 1.

Lines 161-162: If you are using only 40 simulations, why do you explain all this? It is confusing and unnecessary.

The methods section has been significantly reworked to better explain the ensembles that were run representing 1) 81 ensemble members of VolMIP sampled initial conditions and 2) 50 randomly sampled initial conditions. When possible additional information has been removed for clarity.

Line 163: what is NINT?

NINT is defined a few sentences earlier "The atmosphere is represented with non-interactive (NINT) aerosols."

Line 173-174: unclear, the term historical with respect to control is used to describe transient simulations versus unperturbed simulations, so it highlights the type of forcing used (variable, constant). The point is that paired anomalies are calculated as "step-by-step" differences, so they do not include the effect of ongoing unperturbed (or otherwise forced) variability; anomalies from climatologies are instead deviations from a time average (in this case still from the control run), so include ongoing variability. You may also refer to Zanchettin et al. (2022).

This paragraph has been rewritten (now under section 2.3) and now defines explicitly the two types of anomalies examined in the paper: paired anomalies and climatological anomalies. We also include reference to the Zanchettin et al. 2022 paper which discusses paired anomalies for their multi-model ensemble analysis.

Line 181: How GISS model compares with other models is also described in detail by Zanchettin et al. (2022), based on a subset of the simulations used here, see their section 4.2. There, GISS showed some distinct characteristics compared to other models. This should be reported here. Besides, as the author used paired anomalies (as also used in Zanchettin et al., 2022, but for other analyses, not for the radiative flux anomalies) it would be interesting to discuss how this affects the estimation of uncertainties in radiative imbalances. The ensemble spread is quite small in your calculations.

In the revised version of the manuscript, we have moved the detailed results of radiative anomalies into the supplemental materials to make results more concise. We have, however, added reference to the distinct characteristics of radiative Zanchettin et al.

In particular, in section 3 we have added the following description: "In comparison to other models in VolMIP, we note that GISS E 2.1 does display a faster increase of radiative anomalies (Zanchettin et al.). However, between our different ensemble members, there is a little variation in the evolution of the radiative response to the prescribed volcanic forcing (see Figure S1)."

Section 3.1: please provide the direction of all changes (upward or downward). I guess Figure S2 right is for downward flux?

These results have been removed to the supplemental materials but results now include additional information for the forcing diagnostics specifying if they are defined as shortwave or longwave forcing at the surface or top of the atmosphere in the model.

Line 202: normal conditions are not an anomaly of zero, but a range around zero.

This has been corrected to read "mean with an ensemble spread around zero"

Figure 1: please check label (°C)

Figure 1 has been moved to the supplemental materials but now includes correct labels.

Section 3.4.1: how is the NAO index defined? Is it the same as the VolMIP protocol? Anyway, this must be reported. Also, I recommend standardizing the index (for instance using mean and variance of the control run), so that the shown changes can be expressed in terms of standard deviations, so relative to the variability of the index.

The NAO index is defined the same as the VolMIP protocol. While we don't describe details of the index we do refer the reader both to the 2016 VolMIP paper, and to the paper which describes the 500mb geopotential height based index (Stephenson et al. 2006) with the detailed definition in section 1.2.

Line 256: the neutral NAO ensemble is one

This sentence has been removed when updating the discussion of the NAO response for clarity.

Figure 4 and associated text: MSU data are observational, as far as I know. Where are these shown in Figure 4? Why is an observational dataset brought into analysis at this point, whereas for all other analyses there is no such comparison? Volc-pinatubo are idealized experiments. Of course, model-data comparisons are possible, but should be presented and discussed properly.

We apologize for any miscommunication in the presentation of the MSU results. The MSU temperature in the lower stratosphere presented in Figure 4 (now Figure 5) represents the

modelled MSU Temperature metric, which provides a comparable metric to the observations that would be generated for MSU Temperature by satellites. Because we focus this particular figure on the anomaly of the NAO ensemble groups from unperturbed conditions, we do not include a comparison to observations here. We have further clarified the fact that this definition is in fact model-derived and not observational in section 3.2.3.

Line 281: please report p value. The correlation in the right panel seems to be largely determined by a stratification across sub-ensembles based on initial conditions.

In this figure we include a R2 statistica rather than a p-value statistic because we aim to show the relationship between a) vortex strength and the temperature gradient and b) vortex strength and the winter warming anomaly in a way that is comparable to the methodology in Polvani et al. 2019.

To supplement the R2 values which represent the correlation between variables in this figure, we also calculated p values that show the significance of each metric between NAO ensembles, which are reported below as well as in the main text. We also present the p-value for the winter warming response in section 3.4.2.

DJF 1st winter vortex strength: p-value: 2.15e-12

DJF 1st winter temperature gradient : p-value: 9.68e-08

DJF 1st winter warming: p-value: 1.62e-08

Line 511: the paper has been published: https://doi.org/10.5194/gmd-15-2265-2022

The reference for this paper has been updated with the recent publication.

Line 283: ensembles à realizations

I am not clear about what is meant by this comment, but am happy to correct it with further clarification.

Line 297: which historical conditions? Please be detailed in the description of the data that are used (as I suggest above, please add a section on data processing and associated terminology). I would avoid "historical" in this context and use paired anomalies and deviations from climatology. For instance, I am certain most readers would misunderstand the statement at line 393 as well.

What was previously referred to as 'historical anomalies' in the paper is now re-written to be climatological anomalies (as first defined in Methods section 1.3). Hopefully this clarifies the methodology for presented results.

Figure 7: what are the percentiles shown in the box-whisker plots?

The percentiles are the standard box and whisker plot quartiles (25%, 50% 75%)

Line 305-306: this sentence has no meaning to me. I understand what you want to say, but it is not what is read.

This has been re-written to better contextualize with the chosen anomalies and hopefully now is clearer reading "Climatological anomalies show no significant forced response, contrary to the paired anomalies for +NAO and -NAO groups. This suggests that paired anomalies are influenced by the sampled conditions in the unperturbed control. These sampled states of NAO are evident as paired anomalies show cooler than average conditions for - NAO ensembles and warmer than average conditions for + NAO ensembles."

Line 315-319: these sentences are also hard to read and understand… I recommend rewriting this part.

This part has also been re-written with specific reference to the choice of paired anomalies in this probability.

Toohey M, Krüger K, Bittner M, Timmreck C, Schmidt H. The impact of volcanic aerosol on the Northern Hemisphere stratospheric polar vortex: mechanisms and sensitivity to forcing structure. Atmos Chem Phys. 2014;14:13063–79. doi:10.5194/acp-14-13063-2014.
Citation: https://doi.org/10.5194/acp-2023-54-RC1

**RC2: Anonymous**

We thank reviewer 2 for their insightful comments to our manuscript. We hope that our new version of the manuscript will provide a higher level of scientific value both by clarifying our methodology and by more closely discussing differences in the analyzed ensemble response between different sampling schemes and anomalies. We have also conducted some additional analysis in response to their comments which are now incorporated as part of the Supplemental Materials of the manuscript.

This manuscript investigated the impacts of background ENSO and NAO conditions on the responses to Pinatubo-like forcing. Specifically, the authors focus on paired anomalies of different conditions and show that the winter warming can be found with the paired anomalies. The research topic is interesting and crucial, but the provided evidence is not precise enough and several critical scientific issues are not addressed. Therefore, I do not suggest this manuscript to be published in Atmospheric Chemistry and Physics before the authors revise the manuscript to clarify their points and to discuss the issues more comprehensively.

Major comments:

1.  The results in this manuscript highly depend on the paired anomalies, and their definition and interpretation are not well-discussed. The authors should provide more guidance on how to interpret the paired anomalies and why the results are different from the anomalies calculated with the control.

    The new manuscript aims to clarify the definition of anomalies used in this paper (paired and climate anomalies) through re-writing of section 2.3 "Data Analysis and Anomalies", where we discuss not only the definition of anomalies but also point out how they differ in how they include or exclude natural climate variability in the response.

2.  For the discussion related to the polar vortex and winter warming, the authors should check the histogram of the polar vortex strength since the perturbed +NAO and -NAO ensembles have a mean value close to climatology. This may indicate that the NAO state does not have a significant impact when imposing volcanic forcing. The authors should compare the histogram of the control and perturbed 81 ensembles. If the histogram looks similar, it should be considered that there is no significant impact from the chosen NAO states for this model. This is also an issue for the winter warming part.

    We thank the reviewer for their recommendation that we look at the histogram of how NAO conditions are represented in the VolMIP ensemble. To consider this, we looked at the histogram of the Nino 3.4 and NAO Index over both the 81 VolMIP sampled runs and the 50 randomly sampled runs (shown below.) The results shown below suggest

that there are key differences between the control and perturbed distributions in both the ENSO and NAO index. These histograms also illustrate the fact that compared to the Random samples, VolMIP samples show more samples with high or low ENSO and NAO conditions in the control period relative to the perturbed.

We do not include these histograms in the main text but have added them to the supplementals with further description and reference in the main text.

[Figure]

3. The discussions of control and perturbed NAO and ENSO ensembles are, in general confusing. Since the authors use "positive NAO ensemble" but do not say whether this is the volcanic forced positive NAO ensemble or the control positive NAO ensemble. The

authors should make the description more intuitive and easier to follow in order to precisely deliver their arguments.

We thank reviewer 2 for pointing out the confusion in how we define anomalies. We now present further information on both the definition of anomalies (section 1.3) and specifically how we refer to different ensemble groupings such as "positive NAO ensembles" in section 1.2 And 1.3. We hope that this additional information, as well as improving consistency of nomenclature throughout the text adequately clarified the choice of anomalies.

Detail comments:

1. Line 8, "pair anomalies" needs to be explained.

   Section 1.3 now includes a more directed and specific definition of paired and climatological anomalies presented in the paper.

2. Line 9, "winter warming" of what?

   Thank you for pointing this out, winter warming is now defined in the abstract as "warming of Northern Eurasia surface air temperature in the first winter after a volcanic eruption"

3. Lines 12-13, what does it mean for 'relax ENSO anomaly'?

   Thanks again for pointing out this lack of definition in the abstract, it now reads as "... using paired anomalies, we also observe that positive and negative ENSO ensembles tend to decrease tropical sea surface temperature toward baseline conditions"

4. Lines 22-24, any reference for this?
   Additional references for this background information have been added.

5. Lines 54-55, there are papers using large ensemble to study volcanic impact, such as Zanchettin et al., (2022). Please include them and discuss the significance of this manuscript.

   Zanchettin, D., Timmreck, C., Khodri, M., Schmidt, A., Toohey, M., Abe, M., ... & Weierbach, H. (2022). Effects of forcing differences and initial conditions on inter-model agreement in the VolMIP volc-pinatubo-full experiment. Geoscientific Model Development, 15(5), 2265-2292.

   This section of the introduction, "Initial Conditions and Volcanic Eruptions" has been significantly re-written to better contextualize previous work looking at large ensembles

and how initial conditions may impact the modelled response to volcanic eruptions including the reference above. (Lines 440-63)

6. Lines 77-78, there are possibilities of not having El Niño response in different models, such as the aerosol distribution (Ward et al., 2021). Please includes more details of the possibilities.

Ward, B., Pausata, F. S., & Maher, N. (2021). The sensitivity of the ENSO to volcanic aerosol spatial distribution in the MPI large ensemble. In open review for ESD. *Earth System Dynamics*, *12*, 975-996.

Section 1.3 of the introduction "ENSO Response" has also been significantly re-written to include more context of existing studies which have analyzed variability in the ENSO response, and drivers of this variability including aerosol spatial distribution. Several new references have been added including Ward et al. 2021.

7. Lines 97-98, there should be more recent modeling studies for reference.

Lines refer to some common changes in circulation seen both in observations and models after the Mt. Pinatubo eruption. We include some of the early papers that found these results after the Pinatubo eruption which have been replicated by other studies, but choose to leave reference to papers which first brought up these results as a main finding.

8. Lines 180 and 183, please use the same format for the unit.

Thanks for catching this formatting inconsistency. These results have been moved into supplemental materials but now show consistency in their formatting.

9. Line 217, Khodri et al. (2017) uses relative Niño3.4 to indicate the El Niño signal, as so does some others. Please also discuss whether the El Niño signature also does not exist when considering the relative Niño3.4.

For consistency with other studies we have also calculated the Relative SST, the figure of the ENSO response relative to the average SST over the Nino 3.4 region is displayed below with equivalent processing and confidence intervals. Our analysis of RSSTs

suggests that there is little difference between the RSST between the Nino 3.4 [5S-5N] and tropical sst [20S -20 N] between control and perturbed conditions.

While this anomaly is helpful in adding to the discussion of Relative SST response that has been frequently discussed with volcanic eruptions, in the main text we choose to stick with our original Nino 3.4 index with subtracted seasonal average SST over the Nino 3.4 region as it focuses on the climatological of the deviations sst in the Nino 3.4 region. For those who wish to compare, however, we include the relative SST plot in the supplementals and refer to Khodri et al. 2017.

[Figure]

10. Lines 222-223, How do the authors define "greatly" even though clear differences are found between ENSO states?

To clarify this statement, lines 222-223 in the initial manuscript have been replaced with "Overall, all ensembles show post-eruptive cooling of the tropical pacific in the Niño 3.4 region with little difference in the strength of cooling between different initial ENSO conditions. "

11. Line 225, where is the evidence/reference for 'not at all on background NAO phase'?

Thanks for pointing this out. This was meant to point to why we do not look at ENSO response by initial NAO condition in section 3.1, however we do not include figures here as they are ensembles with overlapping spread (no significant difference between NAO initial conditions). This sentence has been deleted from the manuscript for simplification.

12. Line 230, the anomalies in Figure 3 are confusing. Is it a pair-wise anomaly? If yes, please state it; if not, I think the anomaly is not necessary.

Thank you for pointing this out. For simplicity, we leave the definition and discussion of anomaly types used in our analysis in section 1.3 and remove the wording " with the seasonal signal removed" from this line to prevent further confusion. As noted in methods, all anomalies presented are paired anomalies except where climatological anomalies are noted.

13. Lines 230-237, the description is hard to follow, especially for the part for +NAO and -NAO. Does it simply mean that the precondition of NAO does not hold anymore after the volcanic forcing? That is, the precondition of NAO does not impact the volcanic responses in this model.
These lines have now been clarified to talk about +/0/- NAO initial condition groups in terms of mean increases/decreases in geopotential hight relative to mean conditions.

14. Line 238, I cannot infer this argument from Figure 3. If, in total, the histogram really has a reduction of the strong cases, then this argument is valid, but with only Figure 3, this is not the case.
The histogram included earlier and now in the supplementals should now provide further evidence for the reduction in extreme conditions of the NAO.

15. Line 253, 'E).averaging'?

Thanks for catching this formatting issue, the period after parenthesis has now been replaced with a comma.

16. Section 3.4.2, are there corresponding evidence (figures) for the arguments/results?
    Thank you for pointing this out. The reader is now referred to current figure 4 which shows changes in the zonal winds at 10mb and also to a figure in the supplemental figures which shows corresponding changes in geopotential height over the polar region for NAO group means.

17. Lines 255-256, which pattern? And why it can lead to "is driven primarily in pressure changes over the polar region"?

    This line has been removed for clarity.

18. Line 262, (Miller et al.)?

    Parenthetical citation has been changed to a textual citation; thanks for pointing this out.

19. Line 262, Figure 4 should also show the control simulations of +NAO and -NAO.

The new version of this figure includes an additional subplot which also includes the control/unperturbed simulations of all NAO conditions (shown below).

[Figure]

20. Line 265 "north of 60∘"? north of 60°N?
    Yes, this is now corrected.

21. Lines 269-272, I cannot follow whether the authors are discussing the control or the perturbed ensembles. This happens for the entire manuscript.
    In order to clarify our discussion of results we have added additional information in section 1.3 "Data Analysis and Anomalies" Which clearly define what we mean when we discus + ENSO, 0 ENSO, and - ENSO ensembles as used in lines 269. When referring to these ensemble groups we are referring to the anomalous mean response of each of these groups (each with 27 ensemble members.)

22. Line 285, same as previously. The authors need to check whether the anomaly is representable. If the control +NAO has a strong signature, but the perturbed +NAO is close to climatology. Then Figure 6 may be showing the -1*control +NAO signature, meaning that the perturbed +NAO follows the Gaussian distribution and the pre-condition does not change the response of the volcanic eruption.

> All results unless otherwise noted present the paired anomalous response between different ENSO/NAO groups. This is hopefully now clarified in the text. Figure 6 shows areas where there is a statistically different surface temperature response for the ensemble group (NAO condition) from the unperturbed state.

> The perturbed surface temperature response under both +NAO and -NAO conditions does tend towards mean conditions. We discuss this with the winter warming response in figure 7 where we display the perturbed response under each condition rather than the anomalous response. We use figure 6 to discuss the areas where there is a significant anomalous response, to highlight there is a stronger anomalous response in norther Eurasia where studies define the "winter warming response". We do, however, add reference to the fact that +NAO and -NAO ensembles tend toward the displayed neutral NAO conditions.

Citation: https://doi.org/10.5194/acp-2023-54-RC2

---

## Referee Report (RR1)

This manuscript investigated the impacts of initial ENSO and NAO conditions on the responses to Pinatubo-like forcing. Specifically, the authors focus on paired anomalies of different conditions and show that different initial conditions can lead to distinct volcanic responses. The research topic is interesting and crucial, but the manuscript is not well-constructed. Therefore, I do not suggest this manuscript to be published in Atmospheric Chemistry and Physics before the authors revise the manuscript to enhance the readability of the manuscript.

Major comments (The line numbers are referring to the track-change file):

1.  The introduction and model description and experimental setup are still too lengthy and include lots of information that is not related to the study. For example, in 1.2 ENSO response section, the authors should focus only on studies related to tropical eruptions like 1991 Pinatubo. Also, the authors discuss the different mechanisms that cause El Niño responses to volcanic eruptions but none of the mechanisms is mentioned in the results. Similarly, in the Model section, the authors discuss the difference between E2.1G and E2, which is not related to this study.

2.  The discussion includes lots of summary of the results and does not discuss the possibilities of improvement or other points that usually have. The authors should rewrite the discussion section

Other comments:

1.  Line 5-6, possible initial conditions? Why use "possible"? And the entire sentence is difficult to understand

2.  Lines 8-11, this sentence is extremely long and is "with analysis coincident with" grammatically correct?

3.  Line 11, what does it mean for "historical anomalies" and "control conditions"? Does it mean historical and piControl runs or something else?

4.  Line 13, what does it mean for "relax SST toward baseline condition"? baseline condition of what?

5.  Line 16, is "by initial climate conditions present at the time of the volcanic eruption" grammatically correct? And "at the time" is not accurate enough. Do the authors mean the following winter after eruptions?

6.  Line 27, Timmreck et al. (2010)? Please make sure to use the correct format for citations throughout the manuscript. Same issue in Line 30.

7.  Line 29-30, "largest volcanic eruptions in the last decade"? Or the authors mean "been widely studied in the last decade"? It is a bit confusing.

8.  Line 30-32, any reference for this sentence?

9.  Line 32, "a Mt.Pinatubo sized eruption". A volcano can have multiple eruptions. Please use something like "1991 Pinatubo eruption" to indicate which event it is. And this issue happens throughout the manuscript.

10. Line 42, I don't think Zanchettin et al. (2013) discuss the carbon cycle. Please make sure all the references are cited correctly.

11. Line 45, only cite ENSO papers but not NAO papers?

12. Line 154, "The current CMIP6 model of E2.1-G ENSO representation"?

13. Line 158, "Thus we note the model has larger variability in the NAO, likely linked to the model's increased frequency in ENSO events" is this argument in Kelley et al. (2020)? I did not find it.

14. Line 181, "no correlation between ENSO and NAO states"? But in Line 158, the authors mention the NAO variability is related to ENSO events.

15. Line 261, "we do not that"?

16. Line 289, "(Miller et al.)"? "present and equivalent"?

17.

---

## Editor Decision (ED1)

[revised manuscript text omitted]

**Commented [F2]:** If you keep the description of the experiments in the introduction you would maybe need somewhere to add a few sentence explaining the NH winter response.

**Commented [F3]:** Here, I put all the sentences you used to describe your study in the introduction. These need to sorted and maybe somewhat shortened to describe your study.

**Commented [F4]:** Adjust subsection title.

(Adams et al., 2003; Predybaylo et al., 2017; Khodri et al., 2017). This response is suggested to be particularly robust when the eruption occurs in the Northern Hemisphere due to the eruption shifting the ITCZ southward, thus weakening trade winds in the Tropical Pacific (Pausata et al., 2020). Weakened trade winds then cause El Nino like conditions via the Bjerknes feedback (Bjerknes, 1969).

Research has also focused on understanding the dynamics of the El Nino anomaly. One suggested mechanism is the ocean dynamical thermostat (Clement et al., 1996), a mechanism which is suggested to cause advection of warm water through differential cooling. A second hypothesis for a post-eruptive El Nino anomaly is post-eruptive land cooling over tropical Africa which initiates warming through the perturbation of Walker circulation cells (Khodri et al., 2017). This mechanism was also shown to cause a sustained 7-year El Nino anomalies in response to soot aerosols from simulated global nuclear war (Coupe et al., 2021). Predybaylo et al. (2017) and Zambri et al. (2019) additionally studied the robustness of the simulated El Nino anomaly under varying initial conditions at the time of volcanic eruptions. While Predybaylo et al. (2017) found enhanced El Nino like warming for all Mt.Pinatubo simulations except those where eruptions occurred in La Nina years, Zambri et al. (2019) found a consistent warming of tropical sea surface temperature in the Nino 3.4 region of 0.5-1.0°C in response to the 1783 Laki Eruption in the WACCAM model.

Despite several studies supporting El Nino like anomalies, still other observational and modelling studies suggest that there is no statistically significant El Nino like response after several large volcanic eruptions (Dee et al., 2020). These studies argue that anomalies found in observational records and model simulations are not statistically significant, and are rather within the range of natural climate variability (Dee et al., 2020).

**2.3 Northern Hemisphere Winter Response**

The northern hemisphere (NH) experiences a unique response during the first winter after large volcanic eruptions. Many observational (Graf et al., 2007; Christiansen, 2008) and modelling (Timmreck, 2012; Stenchikov et al., 2002) studies have noted a strengthening of the polar vortex the first winter after a large volcanic eruption. This increased polar vortex circulation in the lower stratosphere is closely associated with an enhanced phase of the Arctic Oscillation (AO) and North Atlantic Oscillation (NAO) – two modes of natural climate variability that are separately defined, but closely related in their associated climate impacts including surface temperature patterns (Cohen and Barlow, 2005). Such increased surface temperature patterns have commonly been observed after large volcanic eruptions such as 1991 Mt. Pinatubo (Robock and Mao, 1995; Kelly et al., 1996), and thus the unique signature of increased surface air temperature over Eurasia termed "winter warming" has been analyzed in several volcanic modelling studies.

Modelling studies from previous climate model inter-comparison projects (CMIP) substantiate post-eruptive winter warming. For example (Zambri and Robock, 2016) analyzed an ensemble of CMIP5 simulations finding that most models produce a winter warming signature over the northern hemisphere corresponding with a stronger polar vortex in the lower stratosphere both over historical 1850-2005 simulation period (Zambri and Robock, 2016) and over the last millennium Zambri et al. (2017). Analysis from individual models have also previously supported winter warming corresponding with strengthened polar vortex

Commented [F5]: This paragraph could be shortened. Here it would be enough to only shortly name the processes that cause ENSO anomalies.

circulation: for example the NCAR CAM5 AMIP Large Ensemble showed consistent winter warming in response to both the 1982 El Chinchon and 1991 Pinatubo eruptions (Coupe and Robock, 2021). This increase in surface temperature is also seen in both observational and global modelling studies (Robock and Mao, 1992; Graft et al., 1993).

Still other studies call the robustness of this modelled result into question. For example, other analysis of CMIP5 models show variation in the prevalence of this response (Timmreck et al., 2016; Driscoll et al., 2012) suggesting that large numbers of ensembles may be required to see a significant strengthening of the polar vortex (Bittner et al., 2016). One proposed cause for inconsistencies in the winter warming response is that the simulated winter warming response in a model is within the range of internal variability (Polvani et al., 2019) and thus is not a robust response to volcanic eruptions. Other studies such as Driscoll et al. (2012) and Stenchikov et al. (2006) also find no consistent warming in the northern hemisphere, or strengthening of the polar vortex associated with winter warming.

To better understand why a strengthening of the polar vortex circulation occurs, several studies have proposed mechanisms that link volcanic eruptions with changes in atmospheric circulation Robock and Mao (1995); Robock (2000); Stenchikov et al. (2002). Despite proposed mechanisms, however, some studies suggest that the prevalence of this response may depend on aerosol forcing (Toohey et al., 2014), or may be insignificant in comparison to the range of natural variability in climate (Polvani et al., 2019).

**Commented [F6]:** In each of these subsections of the new subsection2 you could reference to where in the manuscript you discuss the respective processes.

**These text fragments belong rather to method section:**
Further climate modelling experiments have thus been designed to capture variability that may occur due to different initial states of the climate system at the time of a modeled volcanic eruption (Zanchettin et al., 2016). Here, we refer to the states of ENSO and NAO at the time of a prescribed volcanic eruption as "initial conditions" as described by the methodology in the Volcanic Model Intercomparison project (VolMIP, (Zanchettin et al., 2016).

The VolMIP community has looked specifically at how these initial conditions can impact the climate response using a multi-model ensemble, finding minor but significant differences in the climate response to volcanic forcing under different initial conditions (Zanchettin et al., 2022).

---

## Author Response (AR2)

**Dear Editors and Reviewers,**

We thank the editor and two reviewers for their helpful and constructive review of our manuscript. During this review period we have focused on improving our manuscript by restructuring the introduction and discussion, clarifying and correcting statements in the text, and elaborating on the ENSO response by including discussion of the relative sea surface temperature response in comparison to the ensemble in Zanchettin et al. 2022.

It is our hope that the new version of the manuscript improves general readability, enhances comparison with previous studies, and will provide a valuable contribution to the community.

**Sincerely,**

**Helen Weierbach on behalf of all co-authors**

**Dr. Davide Zanchettin**

We thank Dr. Zanchettin for his second review of our manuscript. As he pointed out, clarifying the Nino3.4-RSST response is essential for comparison to both previous and future studies. In response to his comments we have performed further analysis of the Nino 3.4 response using relative sea surface temperature anomalies and also performed a thorough read-through of the text to further polish and clarify the manuscript. Specific responses to Davide's comments are included below.

The authors addressed most of my comments on the original manuscript, and the manuscript has greatly improved, but it seems that there is one point that was not addressed, the one regarding ENSO. In my previous comments I wrote:
"Concerning the analysis of ENSO, the fact that the authors do not identify an El Nino-like response is very likely linked to the fact that the Nino3.4 index "as is" includes the volcanically induced cooling of the whole tropics, which must therefore be removed before investigating dynamical responses of ENSO. The most used approach is based on "relative SST" and is discussed in several papers, for instance Khodri et al. (2017) and Zanchettin et al. (2022). I strongly recommend the authors to revise the ENSO analysis to account for this. Note that using the relative SST method, Zanchettin et al. (2022) report the GISS-E2.1-G "showing a slight warm ENSO anomaly in 1992 in the ensemble-mean", so contrasting the result reported here in this version of the manuscript."
Since the authors still report a La Nina like response, and do not mention the calculation of relative SSTs, I feel my comment was not addressed. As already mentioned, Zanchettin et al., 2022, obtained different results using relative SSTs. I think this is a major point which should be addressed before the manuscript is published.

Thank you for pointing out this point which was not adequately addressed in our original responses. As pointed out, the analysis that we include for the ENSO response during the volcanic simulation which shows cooling of the tropical pacific does in include volcanic cooling of the full tropics. To address this point and enhance comparison between our study in the VolMIP community paper we have completed an analysis of the Nino 3.4 relative sea surface temperature anomalies (RSST- Nino3.4 Anomalies) for all ENSO ensemble groups (figure below with comparison to VolMIP community paper ).

[Figure]

Comparison of ensemble mean
RSST-Nino 3.4 Anomalies for our study (left) and in the VolMIP community paper (right).

For the mean of our full ensemble (n=81), we see RSST anomalies consistent with the Zanchettin et al. community paper where in the first winter, there is consistently a negative anomaly, followed by positive anomalies for some ensembles. This is expected, since the model results used here are an expanded ensemble of the simulations that contributed to the Zanchettin et al. work. Because the RSST-Nino3.4 anomalies do not significantly change the results, we include the RSST-Nino3.4 results in the supplemental materials (S4). Furthermore, we have also decided to include the original seasonally-detrended analysis as it allows us to simultaneously visualize the ensembles for control and perturbed conditions, giving a more easily interpretable representation of how ENSO is changing including the response from volcanic cooling. To discuss the benefits of the two methods, we have additionally added sentences to section 4.1 to provide context and link interested readers to the RSST-Nino3.4 index who are interested.

Also, I have some suggestions for some further polishing of text/clarification that the authors may consider. Below are some examples from abstract, but I encourage the authors to go once more through their manuscript and double check the accuracy of their statements.
Line 3: what is a "regular" time scale? Maybe change to interannual or interannual-to-decadal. Then, it is not the "initial atmospheric and oceanic conditions" that "impact on climate", rather atmospheric and oceanic dynamics contribute to generate intrinsic climate variability, or, initial conditions, or the climate system at a certain point in time, contribute to determine the evolution of climate in the following period.
Line 5: I would rephrase "simulations are sampled from possible initial conditions" as what is done is rather simulations being initialized from sampled conditions
Line 10: I don't understand "with analysis coincident", maybe just "coincident"?
Line 14: "neutral-phase" maybe change to "neutral ENSO". Besides, this sentence reads unclear as it is not reported which sign the SST anomalies are, and if these are the same or opposite sign for positive and negative ENSO ensemble. Maybe it is worth detailing better here.
Line 15: this sentence is unclear, and it reads too vague. That initial conditions affect the response is known, so I would make this final statement more to the point.
Line 332: "show" maybe better "yield"?

We thank Dr. Zanchettin for his thorough comments for clarification from the abstract, it is a big help in finding places to clarify our wording and presentation. All examples included above have been improved and clarified, along with several other sentences throughout the paper.

**Reviewer 2**

We thank reviewer 2 for their thoughtful review of our manuscript. According to their comments we have significantly restructured the introduction, and renamed our discussion section "Discussion and Summary" to better reflect its contents as suggested by the editor.

This manuscript investigated the impacts of initial ENSO and NAO conditions on the responses to Pinatubolike forcing. Specifically, the authors focus on paired anomalies of different conditions and show that different initial conditions can lead to distinct volcanic responses. The research topic is interesting and crucial, but the manuscript is not well-constructed. Therefore, I do not suggest this manuscript to be published in Atmospheric Chemistry and Physics before the authors revise the manuscript to enhance the readability of the manuscript. Major comments (The line numbers are referring to the track-change file):

1. The introduction and model description and experimental setup are still too lengthy and include lots of information that is not related to the study. For example, in 1.2 ENSO response section, the authors should focus only on studies related to tropical eruptions like 1991 Pinatubo. Also, the authors discuss the different mechanisms that cause El Niño responses to volcanic eruptions but none of the mechanisms is mentioned in the results. Similarly, in the Model section, the authors discuss the difference between E2.1G and E2, which is not related to this study.

2. The discussion includes lots of summary of the results and does not discuss the possibilities of improvement or other points that usually have. The authors should rewrite the discussion section

To ensure our new "Discussion and Summary" section includes all relevant information we have added an additional few sentences which now discuss the consequences of our results, and agreement/disagreement with previous studies.

Other comments:

The abstract has been significantly restructured to account for these clarifying comments (1-5) and corresponding comments for the Davide. We hope that the abstract now presents a clearer summary of our study's findings.

1. Line 5-6, possible initial conditions? Why use "possible"? And the entire sentence is difficult to understand

2. Lines 8-11, this sentence is extremely long and is "with analysis coincident with" grammatically correct?

3. Line 11, what does it mean for "historical anomalies" and "control conditions"? Does it mean historical and piControl runs or something else?

4. Line 13, what does it mean for "relax SST toward baseline condition"? baseline condition of what?

5. Line 16, is "by initial climate conditions present at the time of the volcanic eruption" grammatically correct? And "at the time" is not accurate enough. Do the authors mean the following winter after eruptions?

6. Line 27, Timmreck et al. (2010)? Please make sure to use the correct format for citations throughout the manuscript. Same issue in Line 30.

Textual vs. parenthetical citations have been fixed here and throughout the introduction section.

7. Line 29-30, "largest volcanic eruptions in the last decade"? Or the authors mean "been widely studied in the last decade"? It is a bit confusing.

8. Line 30-32, any reference for this sentence?

This sentence was removed/combined with the previous when editing the introduction text.

9. Line 32, "a Mt.Pinatubo sized eruption". A volcano can have multiple eruptions. Please use something like "1991 Pinatubo eruption" to indicate which event it is. And this issue happens throughout the manuscript.

Sentence now reads "a 1991 Pinatubo eruption"

10. Line 42, I don't think Zanchettin et al. (2013) discuss the carbon cycle. Please make sure all the references are cited correctly.

Thanks for catching this. This citation was meant to refer to Zanchettin et al. 2016 and has been corrected. Other citations have also been checked to ensure they refer to the correct literature.

11. Line 45, only cite ENSO papers but not NAO papers?

ENSO citations are now accompanied by two citations to general information on the North Atlantic Oscillation.

12. Line 154, "The current CMIP6 model of E2.1-G ENSO representation"?

This has been changed to "The representation of ENSO in GISS E2.1-G for CMIP6"

13. Line 158, "Thus we note the model has larger variability in the NAO, likely linked to the model's increased frequency in ENSO events" is this argument in Kelley et al. (2020)? I did not find it.

Thanks for pointing this out. The Kelley paper is an incorrect reference here and we have deleted this sentence. There has not (to our knowledge) been any formal discussion about a correlation between increased frequencies of ENSO events and NAO events. We instead only include information about the standard deviation of NAO in the GISS model from Orbe et al. and and corresponding variability in the NAO signal.

14. Line 181, "no correlation between ENSO and NAO states"? But in Line 158, the authors mention the NAO variability is related to ENSO events.

Thank you for pointing this out, as noted in the previous comment we do not have textual evidence of how increased Frequencies of ENSO may affect NAO frequencies. Thus we have removed the previous line and instead only discuss the lack of correlation present in our simulations.

15. Line 261, "we do not that"?

Thanks for pointing out this grammatical error. For some reason, this only exists in the trackchanges document, likely suggesting an error in the latexdiff package that was used to generate the file. The current text does not include this error.

16. Line 289, "(Miller et al.)"? "present and equivalent"?

Citation has been moved to the correct place at the end of the sentence reading "The MSU temperature metric is commonly used as a remotely sensed temperature data metric based on height, however here we present an equivalent modelled metric in E2.1 (Miller et al.)"

**Editor Farahnaz Khosrawi**

We thank Dr. Khosrawi for her very helpful and thoughtful feedback on our manuscript. We have significantly modified the introduction section with the help of her suggestions, renamed and added information to the discussion section, and additionally improved the readability throughout the paper, clarifying several sections throughout.

Dear authors,

please find enclosed two referee reports. While referee one has only minor issues (mainly one comment that has not satisfactorely considered/answered), referee 2 has some major issues on the introduction and discussion. I agree with the referee on the critics on the introduction which is indeed too long and too detailed. It is rather uncommon to have subsections in the introduction and the length should not exceed 1.5 pages.

However, the revision of the introduction can be easily done. I tried to rearange your text (see attached document) and could easily shorten the introduction to a reasonable length. The details on the processes like ENSO, the NH Winter Response could be put in a additional section. With some transitional sentences and references to the respective results sections these text parts could be easily put in a publishable shape.

I personally have no problems with your discussion section. However, to consider the critics of referee 2, one easy solution here would be to simply rename it in "Summary and Discussion" and if possible add some more sentences stating precisly the consequences of your results, agreement/disagreement with previous studies and/or future implications.

Additionally, I also compiled a list with technical corrections as follows:

P2, L27: References should be in parenthesis (\citep instead of \citet).
P2, L52: One closing parenthesis is missing (there is one, but it should be two).
P3, L79: space missing (between text and reference of Coupe et al.).
P4, L98: add "the" -> such as the 1991 Mt Pinatubo.
P4, L104: The Zambri et al. reference should be in parenthesis (\citep instead of \citet).
P4, L117: Same here with the references.
P5, L141: "degree" should be replaced by a degree sign.
P5, L141: "more" appears twice, one is obsolete and should be deleted.
P6, L156: spaces are missing after "December" and "January".
P6, L159: space missing between number and unit.
P6, L160: Something missing here? Should it read "is done with" and is "as for other models"? Generally, the sentence seems to be not entirely grammatically correct. Please check.
P6, L173: add "the" -> the VolMIP protocol.
P6, L179: add/remove space around "=" so that spaces are used equally and use a mathematical "-".
P8, L199: "…..and report and p-value….." -> check sentence, something is wrong here.

P9, Figure 2 caption: Monthly -> monthly, Index -> index

P10, L239: something missing here? Significant? Should it read statistically significant? Otherwise it should read "statistic signal".

P10, L242: Add comma before and after "however" and add "do" so that it reads "we do not do that…..".

P10, L243: Second closing parenthesis missing.

P11, Figure 3 caption: 12:14 -> 12-14?

P11, L267: Temperature -> temperature

P12, Figure 3 caption: Equator-to-pole -> equator-to-pole and remove space between latitude and degree sign.

P12, L268: and -> an?

P12, L270ff: check appearance of degree sign. Either there is a space obsolete or it is written as "degree" instead of a sign.

P14, L301: …..conditions trend towards….. -> should it read "…..conditions have a trend towards…"

P15, Figure 7 caption: Boreal Winter Warming in small letters -> boreal winter warming.

P17, L373: Sentence incomplete?

P17, L379: What are "NINT aerosols"? For what is the abbreviation NINT standing for?

P18, L396: an -> a

Best regards, Farahnaz Khosrawi